# Genetic Disruption of Guanylyl Cyclase/Natriuretic Peptide Receptor-A Triggers Differential Cardiac Fibrosis and Disorders in Male and Female Mutant Mice: Role of TGF-β1/SMAD Signaling Pathway

**DOI:** 10.3390/ijms231911487

**Published:** 2022-09-29

**Authors:** Umadevi Subramanian, Chandramohan Ramasamy, Samivel Ramachandran, Joshua M. Oakes, Jason D. Gardner, Kailash N. Pandey

**Affiliations:** 1Department of Physiology, Tulane University Health Sciences Center, School of Medicine, New Orleans, LA 70112, USA; 2Department of Physiology, Louisiana State University Health Sciences Center, New Orleans, LA 70112, USA

**Keywords:** natriuretic peptides, guanylyl cyclase receptor, cardiac fibrosis, GW788388, TGF-β1, SMAD

## Abstract

The global targeted disruption of the natriuretic peptide receptor-A (NPRA) gene (*Npr1*) in mice provokes hypertension and cardiovascular dysfunction. The objective of this study was to determine the mechanisms regulating the development of cardiac fibrosis and dysfunction in *Npr1* mutant mice. *Npr1* knockout (*Npr1*^−/−^, 0-copy), heterozygous (*Npr1*^+/−^, 1-copy), and wild-type (*Npr1*^+/+^, 2-copy) mice were treated with the transforming growth factor (TGF)-β1 receptor (TGF-β1R) antagonist GW788388 (2 µg/g body weight/day; ip) for 28 days. Hearts were isolated and used for real-time quantitative reverse transcription polymerase chain reaction (qRT-PCR), Western blot, and immunohistochemical analyses. The *Npr1*^−/−^ (0-copy) mice showed a 6-fold induction of cardiac fibrosis and dysfunction with markedly induced expressions of collagen-1α (3.8-fold), monocyte chemoattractant protein (3.7-fold), connective tissue growth factor (CTGF, 5.3-fold), α-smooth muscle actin (α-SMA, 6.1-fold), TGF-βRI (4.3-fold), TGF-βRII (4.7-fold), and phosphorylated small mothers against decapentaplegic (pSMAD) proteins, including pSMAD-2 (3.2-fold) and pSMAD-3 (3.7-fold), compared with wild-type mice. The expressions of phosphorylated extracellular-regulated kinase ERK1/2 (pERK1/2), matrix metalloproteinases-2, -9, (MMP-2, -9), and proliferating cell nuclear antigen (PCNA) were also significantly upregulated in *Npr1* 0-copy mice. The treatment of mutant mice with GW788388 significantly blocked the expression of fibrotic markers, SMAD proteins, MMPs, and PCNA compared with the vehicle-treated control mice. The treatment with GW788388 significantly prevented cardiac dysfunctions in a sex-dependent manner in *Npr1* 0-copy and 1-copy mutant mice. The results suggest that the development of cardiac fibrosis and dysfunction in mutant mice is predominantly regulated through the TGF-β1-mediated SMAD-dependent pathway.

## 1. Introduction

Guanylyl cyclase/natriuretic peptide receptor-A (GC-A/NPRA) is a major biologically active natriuretic peptide (NP) receptor that synthesizes intracellular second-messenger cGMP in response to hormone binding to this receptor protein. Atrial and brain natriuretic peptides (ANP and BNP) are the major cardiac peptide hormones responsible for natriuresis, diuresis, vasorelaxant, and antiproliferative responses, all of which lead to a reduction in blood pressure (BP) and fluid volume homeostasis [1,2,3,4,5]. Mice carrying the global targeted disruption of *Npr1* (encoding NPRA) exhibit a greatly increased incidence of congestive heart failure (CHF) and mortality [6,7,8,9,10,11,12,13]. Our previous studies have demonstrated that global *Npr1* gene disruption in mice induces the expression of hypertrophic markers and matrix metalloproteinases (MMPs) through the activation of a major signaling molecule, the transcription factor nuclear factor-kappa B (NF-κB) [7,10,12,14]. It was also demonstrated that the partial or complete global ablation of *Npr1* promotes cardiac and renal proinflammatory cytokines in vivo [10,15,16,17,18]. 

Our previous studies have identified some of the mechanisms regulating cardiovascular dysfunction in global *Npr1*-ablated mice, demonstrating that *Npr1* disruption upregulates the components of the renin–angiotensin–aldosterone system (RAAS), including increased levels of pro-renin receptor (PRR), angiotensin converting enzyme 1 (ACE 1), angiotensin II (Ang II) type 1 receptor (AT1R), and plasma aldosterone in mutant animals [19,20,21,22]. In contrast, the gene duplication of *Npr1* exhibited anti-inflammatory and antihypertensive effects by reducing the activation and inhibiting the expression levels of the components of RAAS and proinflammatory cytokines [18,19,21,22]. Furthermore, renal immunogenic responses were initiated in *Npr1* gene-disrupted mice such as higher BP, increased levels of Toll-like receptors (TLR2/TLR4) and proinflammatory cytokines, and drastic reductions in regulatory T cells (Tregs) [23,24,25]. In addition, *Npr1* gene-ablated mice showed significantly increased levels of cardiac hypertrophic markers, including beta-myosin heavy chain (β-MHC), proinflammatory mediator NF-κB, and matrix proteins, compared with wild-type (WT) mice, confirming the importance of the *Npr1* in antagonizing hypertrophic disorders [7,10,11,14,19,24,26]. However, the molecular mechanism whereby the ANP/NPRA system exerts its protective effects that mitigate cardiac fibrotic and hypertrophic remodeling in disease states is not well understood. 

A number of fibrogenic factors seem to regulate cardiomyocyte growth, fibroblast activation, and extracellular matrix (ECM) deposition. TGF-β1 is a major profibrotic agent, which drives cardiac cellular signaling in hypertrophic and fibrotic conditions and is known to exert pleiotropic effects in the regulation of critical cellular and biological processes [27,28,29]. The members of the TGF-β1 superfamily influence the diverse nature of cellular processes, including cell proliferation and differentiation as well as programmed cell apoptosis and death [30,31,32,33,34]. Previous reports have shown that the augmented expression of TGF-β1 induces pathological cardiac hypertrophy in cardiac myocytes in response to hypertrophic stimuli, which in turn stimulate myofibroblast transformation and ECM production and deposition [35,36]. Previously, we and others have shown that the augmented expression of TGF-β1 occurs in the pathological hypertrophy induced by *Npr1* gene disruption [7,10,13]. On the other hand, our in vitro studies have shown that TGF-β1 stimulates the transcription factor delta-crystallin enhancer-binding factor 1 (delta-EF-1), which repressed the transcription and expression of *Npr1* in male rat primary vascular smooth muscle cells (VSMCs) and mouse mesangial cells (MMCs) in a dose- and time-dependent manner [37]. In a classical pathway, TGF-β1 interacts with a complex of heterotetrameric transmembrane serine/threonine kinase receptors, namely TGF-β1 receptor type-1 (TGF-β1R1) and type-2 (TGF-β1RII), and leads to the phosphorylation of transcription factors and small mothers against decapentaplegic (SMAD2/SMAD3) molecules, which form a complex with SMAD4 [38]. The heteromeric SMAD complex translocates and enters the nucleus, modulating several other target genes by directly binding to DNA and interacting with promoter-specific transcription factors and coactivators [29]. However, little is known about the mechanisms whereby *Npr1* gene-disruption induces cardiac fibrosis and disorders in disease states. In the present study, we utilized the next-generation TGF-β1 RI and RII antagonist, 4-(4-[3-pyridin-2-yl)-1H-pyrazol-4-yl] pyridin-2-yl)-N-(tetrahydro-2Hpyran-4-yl) benzamide (GW788388), and elucidated the pathological role and molecular mechanisms of TGF-β1/SMAD signaling in cardiac fibrosis induced by global *Npr1* gene disruption in mouse models, including in both male and female animals.

The molecular mechanisms underlying the sex-specific differences in cardiovascular diseases (CVDs) are not well explored. Importantly, male mice with the genetic disruption of *Npr1* are subject to both hypertension and sudden death due to CHF after they reach six months of age, whereas female *Npr1* mice exhibit slightly lower degrees of hypertension but do not suffer sudden CHF and death at adult age [5,7,9,13,21,22,39]. However, most of these cardiovascular mechanisms and dysfunctions have not yet been specifically investigated in a sex-dependent manner [9,10,21,23]. We speculate that fibrotic markers may have critical roles in the onset of cardiovascular dysfunction to sex-specific differences in *Npr1* gene-targeted mutant mice; however, the mechanisms that divergently affect the pathways to cause cardiovascular dysfunction in association with hypertension in female animals remain unclear. To examine these points, we used both male and female global *Npr1* gene knockout mice, which were treated with GW788388 that inhibited TGF-β1 signaling in the mutant animals in the work reported herein.

## 2. Results

### 2.1. Disruption of Npr1 Triggers the Enhanced mRNA Expression of Cardiac Fibrotic Markers and TGF-β1 Receptors 

In order to reveal the mechanisms whereby *Npr1* gene disruption induces cardiac fibrosis, we first attempted to determine the mRNA expression of the fibrotic markers in heart tissues of *Npr1^−/−^* mice by qRT-PCR (Figure 1). Gene-disrupted (*Npr1^−/−^*) as well as heterozygous 1-copy (*Npr1^+/−^*) mice exhibited significantly increased expression levels of cardiac mRNAs of fibrotic marker genes, including collagen-1α (Col-1α; 3.8-fold in 0-copy and 3.2-fold in 1-copy, *p* < 0.001), Col-3 (3.1-fold in 0-copy and 2.5-fold in 1-copy, *p* < 0.001), α-smooth muscle actin (α-SMA) (6.1-fold in 0-copy and 4.9-fold in 1-copy, *p* < 0.001), monocyte chemoattractant protein-1 (MCP-1; 3.7-fold in 0-copy and 3.1-fold in 1-copy, *p* < 0.001), connective tissue growth factor (CTGF; 5.3-fold in 0-copy and 4.6-fold in 1-copy, *p* < 0.001), and plasminogen activator inhibitor-1 (PAI-1; 3.8-fold in 0-copy and 3.6-fold in 1-copy, *p* < 0.001). TGF-β1RI (4.3-fold in 0-copy and 3.6-fold in 1-copy, *p* < 0.001) and TGF-β1RII (4.7-fold in 0-copy and 4-fold in 1-copy, *p* < 0.001), respectively, compared with age-matched 2-copy (*Npr1^+/+^*) control WT mice (Figure 1A–H). However, after the treatment of mice with the TGF-β1 receptor antagonist GW788388, the expression levels of fibrotic marker genes were significantly reduced in both 0-copy and 1-copy mice compared with vehicle-treated mice. Nevertheless, the GW788388-mediated inhibition of most of the mRNAs was still greater than 0-copy and 1-copy mice than control levels in 2-copy WT mice.

### 2.2. Ablation of Npr1 Induces the Expression of SMAD Group of Proteins and TAK-1 and ERK1/2 Proteins

Initially, at baseline, the levels of SMAD group of proteins (SMAD-1, -2, -3, and -4) were significantly (*p* < 0.001) increased in gene-disrupted *Npr1^−/−^* mice compared with WT *Npr1^+/+^* control mice (Figure 2A–D), while the expression of SMAD-6 was found to be decreased in *Npr1* 0-copy mice compared to 2-copy control mice (Figure 2E). With these observations, we hypothesized that the induction and progression of cardiac fibrosis in the *Npr1* gene-disrupted mice are enhanced through the SMAD-mediated signaling pathways (Figure 2A–E). In addition, the protein level of the fibrotic marker such as α-SMA was markedly increased in 0-copy mice compared to 2-copy control animals (Figure 2F). Further, the expression levels of phosphorylated transforming growth factor-β-activated kinase 1 (pTAK1) and phosphorylated extracellular-regulated kinase-1/2 (pERK-1/2) were also increased in *Npr1^−/−^* mice compared with WT control mice (Figure 2G,H).

### 2.3. Disruption of Npr1 Increases the Levels of MMPs, PCNA, and Fibrotic Markers Proteins 

As determined by Western blot and densitometry analysis, the protein levels of ECM molecules, including MMP-2, -9 (MMP-2, 5.3-fold in 0-copy and 5-fold in 1-copy, *p* < 0.001; MMP-9, 3.9-fold in 0-copy and 3.7-fold in 1-copy, *p* < 0.001), proliferating cell nuclear antigen (PCNA; 3.1-fold in 0-copy and 2.5-fold in 1-copy, *p* < 0.001), CTGF (5.4-fold in 0-copy and 2.8-fold in 1-copy, *p* < 0.001), and α-SMA (3-fold in 0-copy and 2.4-fold in 1-copy, *p* < 0.001), were significantly increased in both 0-copy and 1-copy mutant mice compared with 2-copy WT mice (Figure 3A–J). Treatment with GW788388 significantly reduced the quantitative levels of these fibrotic proteins in all three genotypes, including 0-copy, 1-copy, and 2-copy mice. 

### 2.4. Disruption of Npr1 Increases the Levels of TGF-β1RI and TGF-β1RII

The expression of TGF-β1RI (4.8-fold in 0-copy and 3.1-fold in 1-copy, *p* < 0.001) and TGF-β1RII (2.6-fold in 0-copy and 1.7-fold in 1-copy, *p* < 0.001) were significantly increased in 0-copy and 1-copy mice compared to 2-copy control animals (Figure 4A–D). Treatment with GW788388 significantly reduced the quantitative expression levels of both TGF-β1R1 and TGF-β1RII proteins (Figure 4A–D). The expression of pSMAD2 was increased in 0-copy mice (3.2-fold, *p* < 0.001) and 1-copy mice (1.8-fold, *p* < 0.001), whereas pSMAD-3 showed greater increases in both 0-copy mice (3.7-fold, *p* < 0.001) and 1-copy mice (3.5-fold, *p* < 0.001) compared to 2-copy mice (Figure 4E,F). Densitometry analysis showed that the treatment with GW788388 normalized the levels of these proteins in 0-copy and 1-copy mice to the levels in 2-copy control mice. GW788388 also lowered both pSMAD-2 and pSMAD-3 levels in 2-copy mice compared with untreated controls (Figure 4G,H). 

### 2.5. Disruption of Npr1 Enhances the Immunohistochemical Signals of TGF-β1RI, TGF-β1RII and pSMAD-2 in Heart Tissue 

Immunohistochemical analysis revealed that the immunostaining of the protein levels of TGF-β1 receptors (TGF-β1RI and TGF-β1RII) was significantly increased in cardiac cells of 0-copy (*Npr1^−/−^*) and 1-copy (*Npr1^+/−^*) mice compared with age-matched wild-type (2-copy, *Npr1^+/+^*) control animals (Figure 5A,B). Quantitative analysis showed that the immunostaining of TGF-βRI (2.7-fold in 0-copy and 2.2-fold in 1-copy, *p* < 0.001) and TGF-βRII (6.7-fold in 0-copy and 4.5-fold in 1-copy, *p* < 0.001) levels in the heart tissues of 0-copy and 1-copy mice were significantly increased compared with 2-copy control mice (Figure 5C,D). The treatment with GW788388 normalized the immunostaining of the TGF-βI receptors similarly to the control levels of 2-copy mice. In addition, the immunostaining of phospho-SMAD-2 (pSMAD-2), which serves as a marker for the biological activity of TGF-β1, was significantly increased by 3.2-fold in 0-copy mice and 2.9-fold in 1-copy mice (*p* < 0.001) compared to 2-copy control mice. The inhibition of TGF-β1 receptor activity by GW788388 significantly attenuated the immunostaining levels of pSMAD-2 in both 0-copy and 1-copy and also to some extent in 2-copy WT mice (Figure 5E,F). Consistently with the Western blot analysis, the immunohistochemistry results demonstrated a significant increase in pSMAD-2 in 0-copy gene-disrupted mice. 

### 2.6. Treatment of Npr1^−/−^ Mice with GW788388 Attenuates Systolic BP and Cardiac Dysfunction in a Sex-Dependent Manner

In the present study, a pragmatic approach was used to differentiate the role of sex in controlling the hemodynamic parameters in *Npr1* mutant animals. Hence, mice of both sexes were treated with GW788388 and monitored for systolic BP (SBP), cardiac mass and hypertrophy (Figure 6A–D). The SBP was significantly increased in 0-copy male mice (146 ± 5 mmHg) and 1-copy male mice (128 ± 3 mmHg) compared with 2-copy WT male mice (101 ± 2 mmHg) (Figure 6A). Treatment with GW788388 significantly reduced SBP by almost 23 ± 3 mmHg in 0-copy mice and 14 ± 2 mmHg in 1-copy male mice. On the other hand, SBP in 0-copy female mice (129 ± 4 mmHg) and 1-copy female mice (116 ± 3 mmHg) was only modestly increased compared to 2-copy female mice (95 ± 3 mmHg). Treatment with GW788388 reduced SBP by 18 ± 2 mmHg in 0-copy female mice and by 10 ± 2 mmHg in 1-copy female mice (Figure 6A). However, there was no significant effect of GW788388 on the SBP in 2-copy WT mice. The sex-dependent effects showed significant differences in cardiac hypertrophy between male and female *Npr1^−/−^*, *Npr1^+/−^,* and *Npr1^+/+^* mice. The heart weight/body weight (HW/BW) ratio was significantly increased by almost 60% (global hypertrophy) in adult 0-copy male mice (8.02 ± 0.71) and 48% in 1-copy male mice (6.43 ± 0.65) compared to 2-copy wild-type male mice (4.35 ± 0.42) (Figure 6B). However, the HW/BW ratio was only modestly increased in female mutant mice compared with 2-copy female mice. The treatment with GW788388 did not normalize the HW/BW ratio in 0-copy male mice, however, it almost normalized the HW/BW ratio in 1-copy mice (Figure 6B). Significant increases were also observed in the ratio of left ventricular weight/BW (LVW/BW) in both 0-copy and 1-copy male mice compared with age-matched 2-copy male mice (Figure 6C). Treatment with GW788388 normalized the LVW/BW ratio close to the levels in 2-copy WT male mice. The LVW/BW ratio was also significantly altered in 0-copy female mice compared with 2-copy WT female mice (Figure 6C). To more accurately assess the hypertrophic abnormalities in the mutant animals, we measured the HW to tibia length (HW/TL) ratio in both male and female mice. The HW/TL ratio was significantly increased in 0-copy male mice (32.8 ± 1.2) and 1-copy male mice (23.4 ± 0.8) compared with 2-copy WT male mice (15.1 ± 0.4). Treatment with GW788388 did not normalize the HW/TL ratio to the levels observed in control WT mice (Figure 6D). The HW/TL ratio was only significantly altered in 0-copy female mice; however, in 1-copy mice, it remained at almost similar levels to those of the 2-copy control female mice (Figure 6D). 

### 2.7. Ablation of Npr1 Triggers Interstitial and Perivascular Cardiac Tissue Fibrosis 

We carried out histochemical analyses to elucidate the impact of *Npr1* gene disruption on interstitial and perivascular cardiac fibrosis in male mice (Figure 7A–D). Both 0-copy and 1-copy male mice exhibited increased progressive interstitial cardiac fibrosis, with an increased deposition of interstitial collagen fibers (Figure 7A,B). Furthermore, using the TGF-β1 receptor-specific inhibitor GW788388, we determined whether TGF-β1 plays a role in modifying cardiac fibrosis in *Npr1* mutant mice. Male *Npr1* 0-copy and 1-copy mutant mice treated with GW788388 showed attenuated levels of interstitial cardiac fibrosis by almost 65–70% (*p* < 0.001) in 0-copy and 40–45% (*p* < 0.001) in 1-copy male mice compared with untreated control mice (Figure 7A,B). Moreover, no abnormal accumulation of collagen fibers was noted in the control or GW788388-treated 2-copy male mice. Male *Npr1^−/−^* and *Npr1^+/−^* mice also exhibited cardiac perivascular fibrosis but by almost 60–62% (*p* < 0.001) in 0-copy and 40–50% (*p* < 0.001) in 1-copy mice compared with 2-copy male mice, and treatment with GW788388 reversed these fibrotic signals in the mutant animals (Figure 7C,D). 

### 2.8. Differential Hemodynamic Parameters and Echocardiographic Functional Outcome in Male and Female Mutant Mice

The representative images of the M-mode echocardiogram in the left ventricle of control and GW788388-treated male *Npr1* 0-copy, 1-copy, and 2-copy mice are shown in Figure 8. Longer blue lines indicate left ventricular end dimension-diastolic (LVED-d) and shorter blue lines indicate left ventricular end dimension-systolic (LVED-s) in the echocardiogram images. The results of the M-mode echocardiographic analyses reveal that both 0-copy and 1-copy male mice showed increased LVED-s compared with 2-copy male mice (Figure 9A). After treatment with GW788388, LVED-s was significantly reduced to near-normal values compared with 2-copy control male mice. However, 0-copy female mice exhibited only modest increases in LVED-s levels (Figure 9A). LVED-d was also significantly increased in both 0-copy and 1-copy male mice, and after treatment with GW788388, the values were normalized to control levels as seen in 2-copy male mice (Figure 9B). However, both 0-copy and 1-copy female mice showed only modest differences LVED-d compared with 2-copy wild-type control female animals (Figure 9B). Furthermore, both interventricular septal wall thickness (IVST) and posterior wall thickness (PWT) were significantly increased in 0-copy and 1-copy male and female mice, and GW788388 treatment normalized these values to control levels as in 2-copy wild-type mice (Figure 9C,D). Fractional shortening (FS) was severely compromised in both 0-copy and 1-copy male mice, and after GW788388 treatment, these values were significantly improved, similarly to those of control wild-type male mice (Figure 9E). On the contrary, FS was not significantly affected and remained almost normal in 0-copy and 1-copy female mice (Figure 9E). Heart rate (HR) was severely compromised in 0-copy and 1-copy male mice. Nevertheless, after treatment with GW788388, HR was normalized; however, HR was not significantly affected in either 0-copy or 1-copy female mice (Figure 9F). There seem to be distinct sex-specific differential regulations in the cardiac structure and functions of *Npr1* male and female mutant mice.

## 3. Discussion 

The results of the current study demonstrate that the disruption of *Npr1* induces cardiac fibrosis through the TGF-β1/SMAD-dependent pathway by enhancing the expression of SMAD proteins involved in the canonical cascade. A greater degree of interstitial fibrosis was associated with decreased cardiac function in mutant male mice; however, the nature and the severity of fibrosis may vary with the expression levels of different fibrotic genes [40,41,42,43]. The present results strongly support our notion that *Npr1* disruption triggers the cardiac fibrosis and dysfunction through the TGF-β1-dependent pathway. The collagen isoforms and the nature of their cross-linking all might play an important role in myocardial fibrosis, stiffness, and disorders [44,45]. In our study, cellular cardiac hypertrophy was associated with a decrease in cardiac segmental functions (systolic and diastolic). Furthermore, interstitial abnormalities may contribute to the wasted contraction and LV dysfunction in 0-copy mutant mice. Likewise, TGF-β1, through its downstream mediator specifically involving the phosphorylated SMAD2/3, resulted in the increased production of MMP2 and MMP9, thus increasing interstitial and perivascular fibrosis in the *Npr1* mutant animals. We examined the expression of MMP2 and MMP9, which were significantly upregulated in *Npr1* KO mice; however, in the current studies, we did not examine the expression of TIMPs. In our previous studies, we found that cardiac TIMPs are downregulated in *Npr1* KO mice [7]. We predicted that MMPs seem to continue to degrade the ECM, however, there is more collagen deposition due to a possible positive/negative feedback mechanism, which might operate with the continuous activation of MMPs and degradation of collagen fibers. Thereby, the resulting mechanism might contribute to a more cardiac collagen synthesis and concurrently the activation of MMPs in *Npr1* mutant animals.

The activation of the TGF-β1 signaling pathway has diverse physiological roles, including morphogenesis, cell growth, development, and survival [27,31,32,33,46]. In the heart, TGF-β1 activates fibroblast proliferation, resulting in progressive fibrosis [44,45,47]. We and others have reported that the expression of TGF-β1 is induced in response to developmental cardiac hypertrophy and fibrosis in experimental animal models and in end-stage CHF in humans [10,14,18,48]. Our present findings revealed that TGF-β1 exhibits the most potent impact on activating cardiac fibrotic and stress remodeling, particularly in *Npr1* 0-copy male mice. However, these mutant animals are generally presumed to confer deleterious cardiovascular effects, particularly on cardiac dysfunction, often based on the cardiac disorders with the activation of fibrosis [7,14,20]. It is interesting to note that both the inhibition and progression of cardiac fibrosis were blunted in GW788388-treated *Npr1* mutant mice, which likely reflect the circumscribed effect of GW788388 with a targeted reduction in fibrosis. The fact that both interstitial and perivascular fibrosis declined in response to the treatments with GW788388 indicates that the net effect of *Npr1* gene disruption seems to be mediated via TGF-β1-dependent cascades. 

Our results highlight that the retrospective expression of cardiac fibrotic markers, which are likely to act as candidates of mediators for the fibrosis, such as CTGF and MCP-1; however, CTGF is one of the most prominent factors in the sequence of the cascade of cardiac fibrosis [49,50,51,52]. Here, we show that the ablation of *Npr1* stimulates CTGF expression in the heart tissues, and our findings suggest that cardiac fibrosis correlates with a persistent increase in the expression of TGF-β1RI and TGF-β1RII in the heart tissues of *Npr1* 0-copy and 1-copy mice. However, TGF-β1 receptor activation has been shown to be associated with the activation of pTAK-1 and pERK1/2, which are also stimulated by inflammatory cytokines such as interleukin-1 (IL-1) and tumor necrosis factor-alpha (TNF-α) in the fibrotic pathways [52,53]. Thus, the targeted canonical TGF-β1 signaling cascade might play a central role in the maladaptive cardiac fibrosis and dysfunction in *Npr1^−/−^* mutant mice. 

Our results support the benefits of inhibiting the TGF-β1 pathway utilizing the TGF-β1 receptor antagonist GW788388, however, the inactivation or suppression of the receptor was not sufficient to completely reverse the pathophysiology of cardiac disorders. Hence, alternative pathophysiologic stimuli that can directly or indirectly activate TGF-β1 signaling may also be involved in the activation of fibrosis and cardiac dysfunction mechanisms different from that of the primordial canonical pathway. Thus, additional strategies may be required to optimally attenuate cardiac fibrosis and dysfunction in *Npr1^−/−^* mice. The present results indicate an increased expression of profibrotic markers (Col-1α, MCP-1, CTGF, α-SMA) and ERKs (ERK1/2, pERK1/2), a sequence of events that could readily expand the ECM proteins in the heart tissues, contributing to diastolic dysfunction in these animals. This notion is supported by the findings that the inhibition of TGF-β1 signaling prevented the development of cardiac fibrosis and hypertrophy in the experimental animals [43,44,54,55]. 

We noted that some critical differences were found in the effect of TGF-β1 signaling leading to cardiac hypertrophy and dysfunction between male and female mice. Particularly, the SBP, HW/BW, and HW/TL ratios were significantly different between both sexes of mice with and without the treatments with TGF-β1 receptor antagonist, GW788388. Similarly, the echocardiographic data, including LVED-s, LVED-d, IVST, and PWT were also significantly different between male and female mice, with a greater degree of cardiac dysfunction in males than in females. In the present studies, HR was compromised in mutant *Npr1* male mice; however, it was normalized after treatment with the TGF-β1 receptors antagonist, GW788388. However, HR was not significantly affected in mutant female mice. We speculate that HR seems to be compromised in male mutant mice due to the CHF episode, especially in 0-copy mice, usually after 6 months of age [7,13]. These observations indicated that TGF-β1 exhibits a significant impact on the activating stress remodeling in *Npr1* KO mice in a sex-dependent manner. We speculate that the dimorphic sex-specific differences in *Npr1* KO mice might be to some extent due to protective effects of sex hormone estrogen in female animals. However, the sex-specific differences in these *Npr1* mutant animals need to be investigated in greater detail.

Evidence suggests that ANP-GC-A/NPRA signaling exerts a counter-regulatory effect on the activated TGF-β1-induced signaling cascade [56,57]. Previous studies have also revealed the role of intracellular cGMP as a negative regulator of TGF-β1 signaling and function [58,59,60]. On the other hand, we have reported that TGF-β1 signaling exerted inhibitory effect on the transcriptional expression of *Npr1* in primary MMCs and VSMCs and also inhibited the vasorelaxant reactivity of aortic rings ex vivo involving transcription factor, delta-EF-1 [37]. TGF-β1 stimulation has been shown to increase SMAD-2 and SMAD-3 protein phosphorylation [61,62,63]. The SMAD-2 and -3 proteins are important intracellular molecules, which mediate the cellular signaling of TGF-β1 superfamily ligands [64,65]. Upon phosphorylation, pSMAD-2 and pSMAD-3 are heterodimerized and form complexes with SMAD-4, which translocate to the nucleus and modulate the transcriptional regulation of several genes encoding ECM proteins and fibrotic and hypertrophic markers [66]. GW788388 treatment suppressed the expression of phospho-SMAD-2 and phospho-SMAD-3 in heart tissue, indicating that the SMAD-dependent canonical TGF-β1 signaling plays an important role in cardiac fibrotic crosstalk. Our finding suggests that the TGF-β1 signaling requires the recruitment of SMAD2/SMAD3 to trigger the fibrosis and hypertrophic disorders in *Npr1* KO and HT mutant mice. Hence, our proposed model supports the notion that the observed effect of GW788388 is largely on the SMAD-dependent TGF-β1-mediated signaling pathway in *Npr1* gene-ablated mutant mice (Figure 9). A role for SMAD-2 and SMAD-3 signaling in fibrosis has also been suggested in post-infarction myocardial fibrosis [67,68]. In addition to the canonical signaling pathway, TGF-β1 can also promote non-canonical signaling cascades, including TAK-1, ERK1/2, and NF-kB pathways leading to disease states [69,70]. Usually, TGF-β1 signaling contributes to cardiac disorders with the fibrotic events [36,71]. On the contrary, it has been indicated that the activation of TGF-β1/SMAD signaling repaired ventricular fibrosis after myocardial infarction [72]. Nevertheless, the deficiency of TGF-β1 and its signaling components, including SMAD proteins, triggered the disease conditions such as inflammatory bowel disease and decreased cardiac function [73,74]. Targeting TGF-β1 signaling using the neutralizing antibodies resulted in an enhanced LV dilation and myocardial infraction with increased mortality [71,75]. It has been suggested that the therapeutic usage of antagonists of the TGF-β1 signaling cascade seems to elicit a toxic effect and remains inefficacious in the clinical trials [36]. However, a more specific and effective therapy is not yet available and await further research.

In the present study, we focused on the mechanistic aspects of comparative TGF-β1 signaling cascade in the heart of *Npr1* 0-copy, 1-copy, and 2-copy mice. The results demonstrate that the blockade of TGF-β1/SMAD signaling with the new generation TGF-β1 receptor (TGF-β1R) antagonist, GW788388, dramatically inhibited the cardiac fibrosis and dysfunction in *Npr1* mutant animals. GW788388 antagonized the expression of potential fibrotic markers by inhibiting the expression and recruitment of SMAD proteins necessary for the activation of TGF-β1/TGF-β1RI/II complex and downstream signaling cascades, thus improving cardiac function in both *Npr1* KO and HT mutant mice. The global ablation model of *Npr1* in mice provided a broad support for the studies of the pathophysiology of cardiac fibrosis, hypertrophy, and remodeling in disease states (Figure 10). The mutations or gene-disruption encoding pro-ANP (*Nppa*) and NPRA (*Npr1*) trigger high BP and CVDs in the genetically altered animal models [4,5,8,13,76]. The previous clinical and genetic studies strongly support our present findings that a positive association exists between the polymorphisms of *Nppa*, pro-BNP (*Nppb*), and *Npr1*, causing essential hypertension, increased cardiac mass, and CVD [77,78,79,80,81,82,83,84]. 

In conclusion, our results demonstrate that an increase in cardiac TGF-β1 signaling was accomplished by the increased cardiac SMAD phosphorylation in *Npr1^−/−^* mutant mice. Thus, the pharmacological inhibition of the TGF-β1 in the present study might lead to a feasible strategy for investigating the novel mechanisms of TGF-β1 action in cardiac fibrosis and dysfunction in the absence of ANP/NPRA/cGMP signaling. Overall, the present findings provide new insights into *Npr1* gene-disruption-mediated cardiac fibrosis, hypertrophy, and functional disorders suggesting that the ANP/GC-A/NPRA/cGMP signaling cascade might provide possible molecular protective mechanism for cardiovascular disorders in humans of both sexes.

## 4. Materials and Methods

### 4.1. Materials

The TGF-β1 receptor antagonist GW788388 was obtained from R&D Systems (Minneapolis, MN, USA). RNeasy mini-kit for total RNA isolation, RT^2^ First Strand cDNA kit, and RT² SYBR Green/ ROX Master Mix were obtained from Qiagen (Valencia, CA, USA). Sequence-specific oligonucleotides were purchased from Eurofins MWG (Operon, AL, USA). Primary antibodies for MMP-2, MMP-9, TGF-β1, TGF-β1R1, TGF-β1RII, ERK1/2, and pERK1/2 were obtained from Santa Cruz Biotechnology (San Diego, CA, USA). SMAD-1, -2, -3, -4, and -6 antibodies were purchased from Cell Signaling Technology (Danvers, MA, USA). Similarly, the phosphorylated antibodies for pSMAD2, pSMAD3, and pTAK1 were also received from Cell Signaling Technology. The detailed descriptions of the antibodies are listed in Table 1. SuperSignal West Femto Chemiluminescent Substrate Western blot detection reagent kit was obtained from Thermo Fisher Scientific (Rockford, IL, USA). All other reagents used were of molecular biology and analytical reagent grade.

### 4.2. Generation and Genotyping of Mice 

Global *Npr1* gene-disrupted mice were produced by homologous recombination in embryonic stem cells as previously described [9,85]. The mice were bred and maintained at the animal facility of the Tulane University School of Medicine Vivarium. Animals were handled according to the protocols approved by the Institutional Animal Care and Use Committee. The mouse colonies were housed under 12 h light–12 h dark cycles at 25 °C and fed regular chow (Purina Laboratories, Framingham, MA, USA); tap water was available ad libitum. All animals were littermate progeny of C57/BL6 genetic background and were designated as *Npr1* gene-disrupted homozygous (*Npr1*^−/−^, 0-copy), heterozygous (*Npr1*^+/−^, 1-copy), and wild-type (*Npr1*^+/+^, 2-copy) mice. In this study, adult (14–16 weeks) male and female mice were utilized. The animals were genotyped by PCR analyses of DNA isolated from tail biopsies, and PCR was performed using our standard method as previously published [9,39,85].

### 4.3. Experimental Animal Groups

Adult (14–16 weeks) 0-copy (n = 24), 1-copy (n = 24) and age-matched 2-copy (n = 24) mice were used in the present study and treated with TGF-β1 receptor (TGF-β1R) antagonist GW788388. In preliminary studies, we determined the dose (0.5 to 5 µg/g body weight/day) and durations (4–5 weeks) of treatment for the mice with the intraperitoneal (ip) administration of GW788388. We treated the mice with an optimum dose of GW788388 (2 µg/g body weight/day) for 28 days and found no toxicity of the compound in the treated animals. In previous studies, GW788388 was utilized at 2–5 mg/kg body weight/day for 4–5 weeks of duration [86,87,88]. Mice were divided into two groups: Group I (n = 6), vehicle-treated (control); and Group II (n = 6), GW788388-treated (2 µg/g/day). A stock solution of GW788388 was prepared at 20 mg/mL concentrations in dimethyl sulfoxide (DMSO) diluted in olive oil and stored at −80 °C. On the day of treatment, the drug was thawed, diluted with olive oil to appropriate concentrations, and vortexed for 1 min at room temperature. Mice were administered GW788388, ip, while control groups were injected with vehicle (DMSO and olive oil). At the end of the experiment, blood was collected and the heart was isolated from the treated and control mice for histological, biochemical, and molecular analyses.

### 4.4. Blood Pressure and Cardiac Function Analyses 

SBP, heart rate, and echocardiography measurements were made in control and drug-treated mice, which were trained for one week prior to actual experimental measurements. Cardiac structure and function were assessed via echocardiography using the Vevo 3100 Imaging System with a 30-MHz probe (VisualSonics, Toronto, ON, Canada) as previously described [89]. Mice were anesthetized with 1–1.5% isoflurane and placed onto an integrated heating pad to maintain body temperature. B-mode and M-mode images of the long and short axis were recorded. Images of the left ventricular chamber were analyzed using the leading-edge method in Vevo LAB 5.5.1 Software (VisualSonics, Toronto, Canada). Group averages were calculated from measurements made from a minimum of three cardiac cycles per animal to determine the FS, LVED-s, LVED-d, IVST, and PWT. The readings were taken from the average of six sessions/day for five consecutive days [7,89]. SBP was measured by noninvasive computerized tail-cuff method (Visitech-2000) as previously reported [25,90]. BP was calculated and determined using the average of six sessions per day for seven consecutive days.

### 4.5. Blood and Tissue Collection 

Mice were euthanized by the administration of a high concentration of CO_2_ gas. Blood was collected by cardiac puncture from mice under CO_2_ anesthesia into chilled tubes containing 10 µL of heparin (1000 USP units/mL). Plasma was separated by centrifuging the blood samples at 3000 rpm for 10 min at 4 °C and then stored at −80 °C until used. Heart tissues were dissected, weighed, snap frozen in liquid nitrogen, and stored at −80 °C until used. A portion of heart tissue was sliced, kept in 10% buffered formalin overnight, and processed for histological studies. 

### 4.6. Measurement of Cardiac Hypertrophy and Interstitial Fibrosis 

At the end of the experimental period, the body weights of mice were measured and hearts were dissected out, blotted, and weighed. The ratios of HW/BW, LVW/HW, and HW/TL were calculated as an index of cardiac hypertrophy. Paraffin-embedded tissue sections (5 μm) were stained with Picrosirius red to disclose the presence of accumulated interstitial collagen fibers as a marker of cardiac fibrosis. The ratio of interstitial and perivascular fibrosis to the total left ventricular area was calculated in a blinded and unbiased manner from 20 randomly selected microscopic fields in six sections per heart using ImagePro Plus image analysis software (Media Cybernetics, Silver Spring, MD, USA).

### 4.7. Real-Time RT-PCR Analysis 

Total RNA was extracted using an RNeasy plus mini-kit. Approximately 30 mg of heart tissue was homogenized, and RNA was extracted according to the manufacturer’s instructions (Qiagen, Valencia, CA, USA). First-strand cDNA was synthesized from 1 μg of total RNA in a final volume of 20 μL using an RT^2^ First Strand kit. We performed qRT-PCR using the Mx3000P real-time PCR system. Data were analyzed with MxPro software (Stratagene, La Jolla, CA, USA) as previously described [14]. The forward (F) and reverse (R) primers used are: Col-1α, F-5′-ttctcctggcaaagacggac-3′ and R-5′-ccatcggtcatgctctctcc-3′; Col-3, F-5′-gaggaatgggtggctatccg-3′ and R-5′-ttgcgtccatcaaagcctct-3′; α-SMA, F-5′-gccgagatctcaccgactac-3′ and R-5′-ataggtggtttcgtggatgc-3′; CTGF, F-5′-agcggtgagtccttccaaag-3′ and R-5′-ttcatgatctcgccatcggg-3′; MCP-1, F-5′-caggtccctgtcatgcttct-3′ and R-5′-gtggggcgttaactgcatct-3; PAI-1, F-5′-tcgtggaactg ccctaccag-3′ and R-5′-agacttgtgaagtcggccag-3′; TGF-βRI, F-5′-agctcctcatcgtgttggtg-3′ and R-5′-aaaccgacctttgccaatgc-3′; TGF-βRII, F-5′-acgttcccaagtcggatgtg-3′ and R-5′-ttcagtggatggatggtcct-3′; GAPDH, F-5′-tccctcaagattgtcagcaa-3′ and R-5′-gatccacaaacggatacatt-3′. PCR amplification was carried out in triplicate in a 20 μL reaction volume using RT^2^ real-time™ SYBR Green/ROX PCR Master Mix. The reaction conditions were 95 °C for 10 min, followed by 45 cycles at 95 °C for 15 s and 60 °C for 1 min; this was followed by 1 cycle at 95 °C for 1 min, 55 °C for 30 s, and 95 °C for 30 s for the dissociation curve. The reaction mixture without template cDNA was used as a negative control. Threshold cycle numbers (C_T_) were determined with MxPro QPCR Software and transformed using the ΔC_T_ comparative method. The mRNA expression was normalized to the expression values of glycerolaldehyde phosphatedehydrogenase (GAPDH) as an endogenous control within each sample and relative to positive and negative controls. The level of gene expression was determined by the comparative Ct method (ΔΔC_T_). After PCR amplification, the melting curve of each amplicon was determined to verify its accuracy.

### 4.8. Preparation of Cytosolic and Nuclear Extracts 

Heart tissues were homogenized in an ice-cold 10 mM Tris-HCl buffer (pH 8.0) containing 0.32 M sucrose, 3 mM calcium chloride (CaCl_2_), 2 mM magnesium acetate (MgOAc), 0.1 mM ethylenediaminetetraacetic acid (EDTA), 0.5% Nonidet P-40 (NP-40), 1 mM dithiothreitol (DTT), 0.5 mM phenylmethylsulfonyl fluoride (PMSF), 1 mM orthovanadate, 30 mM sodium fluoride (NaF), and 10.0 µg/mL each of leupeptin, aprotinin, and pepstatin. The homogenate was centrifuged at 8000× *g* at 4 °C for 20 min. The supernatant was separated and stored as a cytosolic fraction. The pellet was washed three times in wash buffer by resuspending it with a 20-gauge needle and centrifuging at 6000× *g*. The pellet was resuspended in a low-salt buffer (20 mM HEPES, pH 7.9; 1.5 mM MgCl_2_; 20 mM KCl; 0.2 mM EDTA; 25% glycerol; 0.5 mM DTT; and 0.5 mM PMSF), incubated on ice for 5 min, and mixed with an equal volume of high-salt buffer containing 20 mM HEPES (pH 7.9), 1.5 mM MgCl_2_, 800 mM KCl, 0.2 mM EDTA, 1% NP-40, 25% glycerol, 0.5 mM DTT, 0.5 mM PMSF, 1 mM orthovanadate, 30 mM NaF, and 10.0 µg/mL each of leupeptin, aprotinin, and pepstatin. The mixture was incubated on ice for 30 min and centrifuged at 14,000× *g* for 20 min. The supernatant was separated and stored as a nuclear fraction at −80 °C until used.

### 4.9. Western Blot Analysis

Tissue homogenate (30 μg of protein) was mixed with an equal volume of sample loading buffer and separated under reducing conditions by 10% sodium dodecyl sulfate-polyacrylamide gel electrophoresis (SDS-PAGE). The separated proteins were transferred to a polyvinylidene difluoride (PVD) membrane at 100 V. The membrane was blocked with 1X Tris-buffered saline/Tween 20 (TBST; pH 7.5) containing 5% nonfat dry milk powder for 1 h at room temperature, then incubated overnight at 4 °C with specific primary antibodies in TBST containing 3% nonfat dry milk powder as previously reported [7,91]. The appropriate dilutions of specific antibodies are listed in Table 1. After three washes with TBST for 5 min each, the membrane was incubated for 1 h in horseradish-peroxidase-conjugated corresponding secondary antibodies at a dilution ratio of 1:5000, washed three times with TBST, and developed using the SuperSignal West Femto Chemiluminescent Substrate Western blot detection reagent kit. The luminescent signal was detected using the Alpha Innotech imaging system (San Loreno, CA, USA). 

### 4.10. Immunohistochemical Analysis 

The heart tissue sections embedded in paraffin were dewaxed and rehydrated using a series of alcohol and TBS solutions. The rehydrated sections were then incubated in citrate buffer (pH 6.0) for antigen retrieval, and the peroxidase was blocked using 0.3% H_2_O_2_ in methanol as previously described [14,25]. Nonspecific binding was blocked using 3% BSA in TBS for 1 h and the sections were incubated with TGF-β1RI and TGF-β1RII specific primary polyclonal antibodies (1:250) diluted in 1% BSA in TBST for 2 h at room temperature. Heart sections were washed with TBST and then incubated with corresponding HRP-labeled secondary antibody for TGF-β1RI and TGF-β1RII at a dilution of 1:2500 for 1 h at room temperature. At the end of the incubation, the slides were washed and the peroxidase activity was visualized by treating the slides with 3,3′-diaminobenzidine (DAB) and counterstained with Meyer’s hematoxylin. Similarly, the negative controls were incubated with TBST instead of primary antibodies. The cells showing positive reactivity to the proteins were calculated from the quantitative analysis under a light microscope using Image-Pro Plus analysis software (Media Cybernetics, Silver Spring, MD) in a blinded and unbiased manner. Each section was examined at high magnification (400×) and the percent of positive area was indicated. 

### 4.11. Statistical Analysis 

The results are expressed as the mean ± S.E. Statistical significance was evaluated by one-way analysis of variance (ANOVA) followed by Dunnett’s multiple comparison tests using GraphPad Prism Software (GraphPad Software, San Diego, CA, USA). A *p*-value of <0.05 was considered significant.

## Figures and Tables

**Figure 1 ijms-23-11487-f001:**
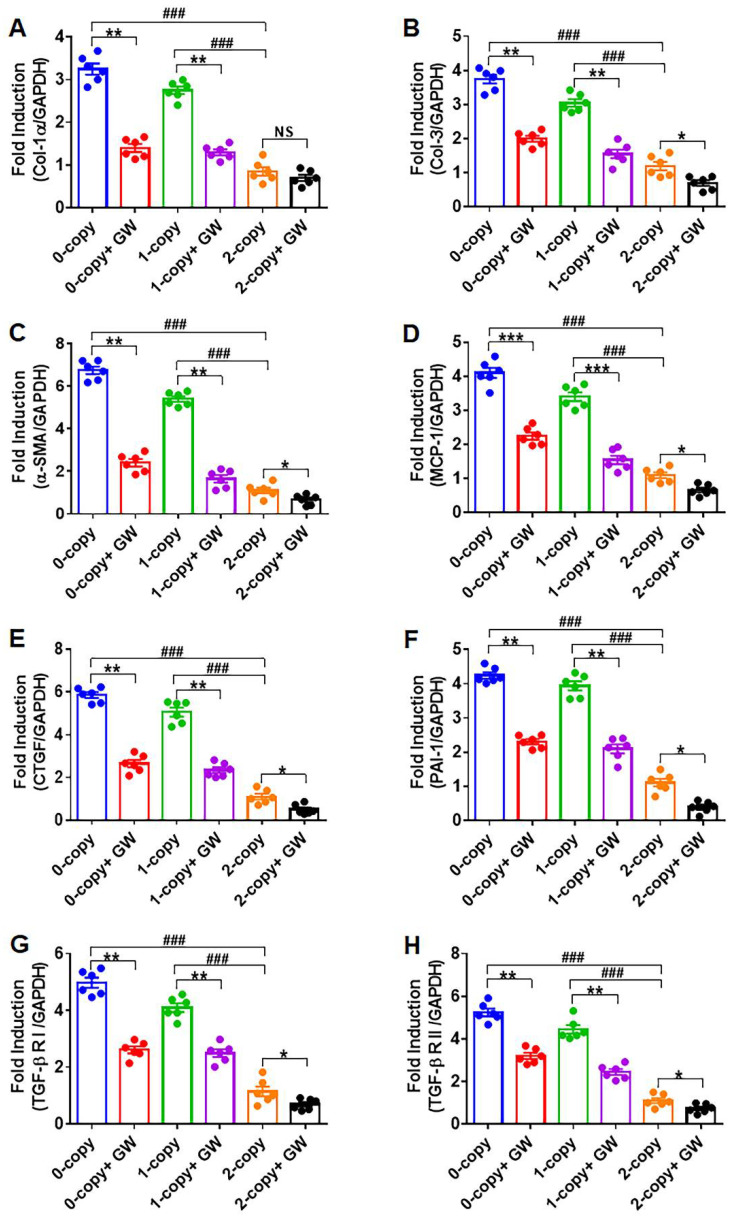
The expression analysis of the mRNA of fibrotic marker genes in the heart tissues of control and GW788388 drug-treated male *Npr1* gene-disrupted mice: The expression levels of mRNA of Col-1-α, Col-3, α-SMA, MCP-1, CTGF, PAI-1, TGF-β1RI, and TGF-β1RII in the heart tissues of control and drug-treated mice are shown in panels (**A**–**H**). The expression of mRNA was normalized to GAPDH expression. Values are expressed as means ± S.E. (n = 6 animals in each group). Statistical significance is expressed as ### *p* < 0.001, vehicle-treated 0-copy and 1-copy vs. 2-copy; * *p* < 0.05, ** *p* < 0.01, *** *p* < 0.001, vehicle-treated 0-copy, 1-copy and 2 copy vs. drug-treated same gene copy number. NS: not significant.

**Figure 2 ijms-23-11487-f002:**
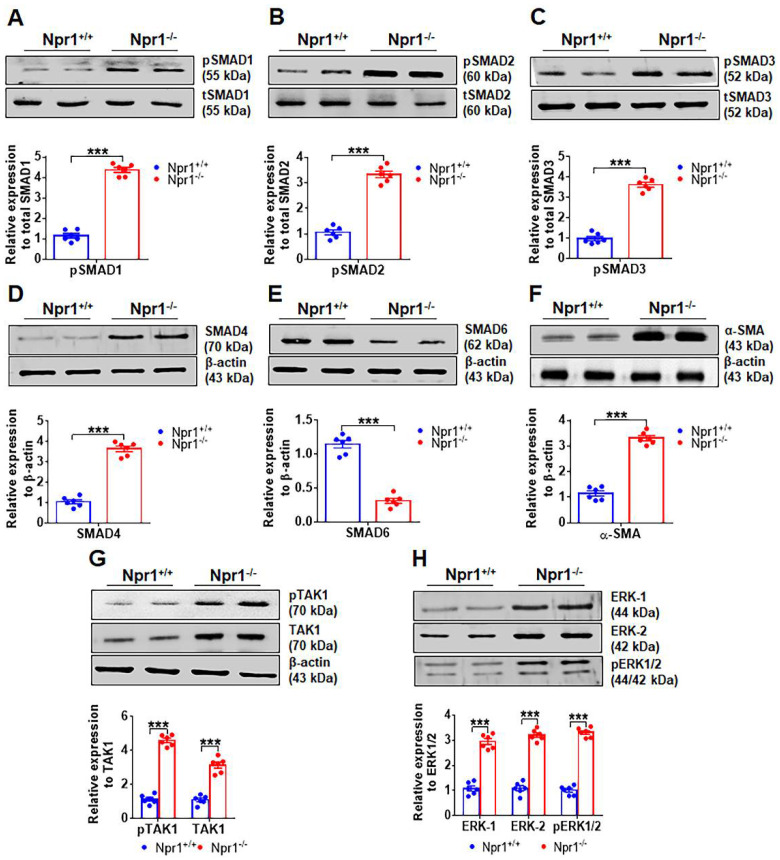
Analysis of the protein levels of different SMAD and key fibrotic markers in the heart tissues of male *Npr1* gene knockout mice: representative Western blots showing the levels of pSMAD-1, pSMAD-2, pSMAD-3, SMAD-4, SMAD-6, α-SMA, pTAK1, and pERK 1/2 proteins in the heart tissues of *Npr1* mice, respectively (panels (**A**–**H**)). The densitometry analysis of proteins was normalized to total SMAD proteins and/or β-actin protein levels. Values are expressed as means ± S.E. (n = 6 animals in each group). Statistical significance is expressed as *** *p* < 0.001, *Npr1*^+/+^ (2-copy) vs. *Npr1*^−/−^ (0-copy). In panels (**A**–**F**), Western blot was performed for the cytosolic extract and in panels (**G**,**H**), whereas Western blot was performed for both cytosolic and nuclear extracts.

**Figure 3 ijms-23-11487-f003:**
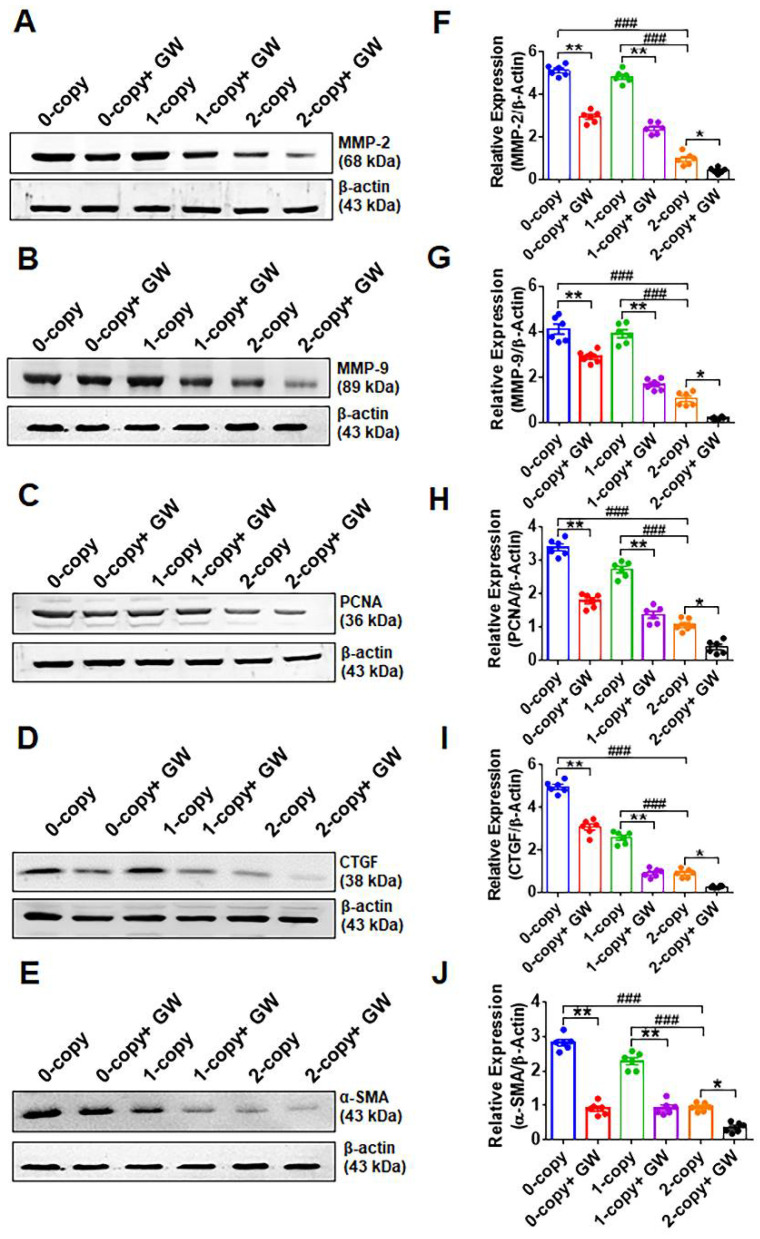
Western blotting analysis of extracellular matrix and fibrotic proteins in the heart tissues of control and drug-treated male *Npr1* gene-disrupted mice: representative Western blots showing the levels of MMP-2, MMP-9, PCNA, CTGF, and α-SMA proteins (panels (**A**–**E**)). The densitometry analysis of proteins was normalized to β-actin protein levels (panels (**F**–**J**). Values are expressed as means ± S.E. (n = 6 animals in each group). Statistical significance is expressed as ### *p* < 0.001, vehicle-treated 0-copy and 1-copy vs. 2-copy; * *p* < 0.05, ** *p* < 0.01, vehicle-treated 0-copy,1-copy and 2 copy vs. drug-treated same gene copy number.

**Figure 4 ijms-23-11487-f004:**
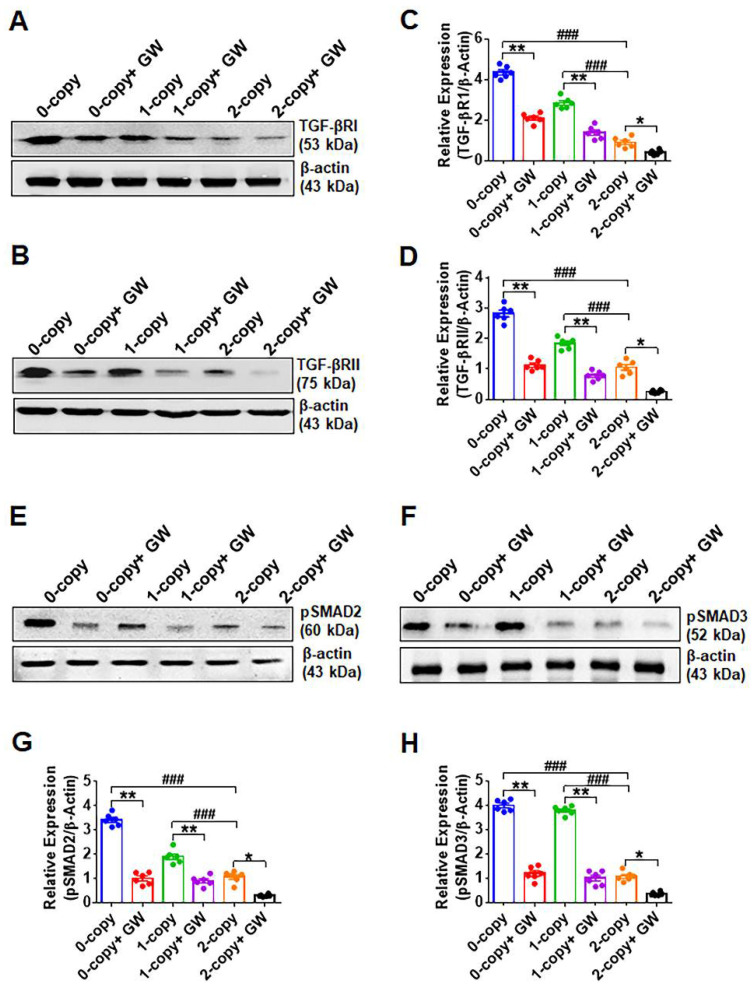
Western blot analysis of TGF-β1 receptors and SMAD proteins in the heart tissues of control and drug-treated male *Npr1* gene-ablated mice: representative Western blots showing the protein expression of TGF-β1RI and TGF-β1RII (panels (**A**,**B**)). The densitometry analysis of proteins was normalized to β-actin expression (panels (**C**,**D**)). Representative Western blots showing the protein levels of pSMAD-2 and pSMAD-3 (panels (**E**,**F**)). Densitometry analysis of pSMAD-2 and pSMAD-3 proteins was normalized to β-actin (panels (**G**,**H**)). Values are expressed as means ± S.E. (n = 6 animals in each group). Statistical significance is expressed as ### *p* < 0.001, vehicle-treated 0-copy and 1-copy vs. 2-copy; * *p* < 0.05, ** *p* < 0.01, vehicle-treated 0-copy,1-copy and 2 copy vs. drug-treated same gene copy number.

**Figure 5 ijms-23-11487-f005:**
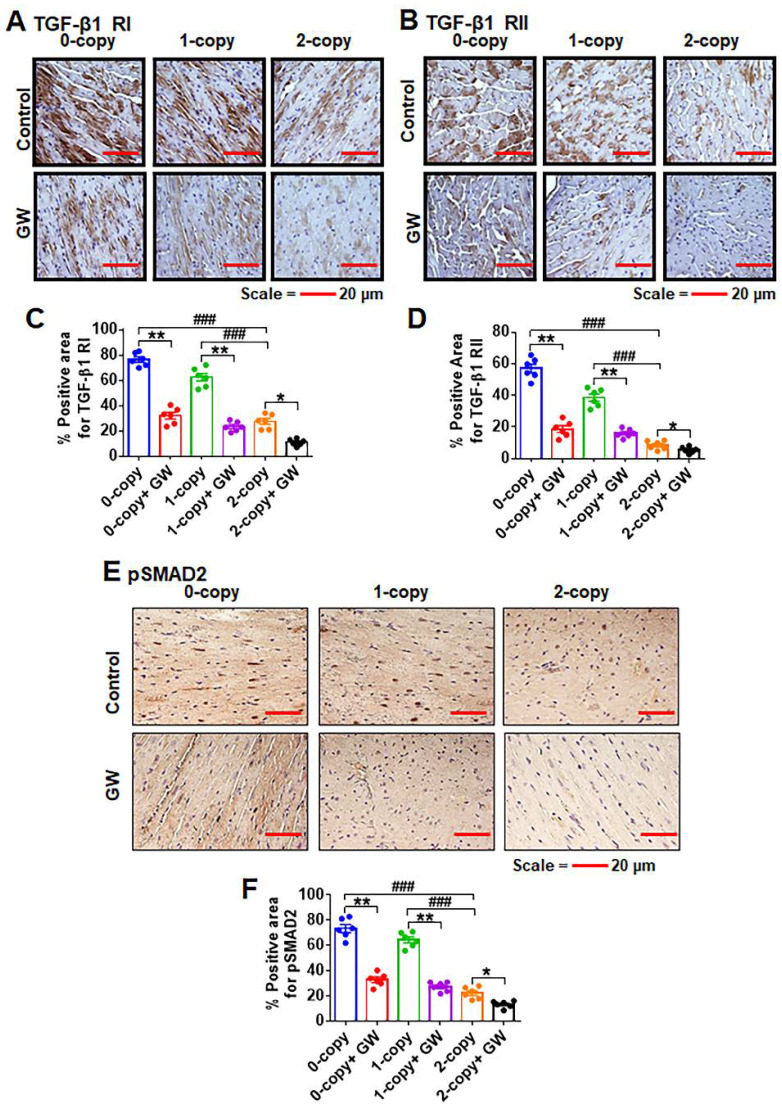
Immunohistochemical analysis of the TGF-β1RI, TGF-β1RII, and pSMAD-2 proteins in heart tissue sections of control and drug-treated male *Npr1* gene knockout mice: representative immunohistochemistry of TGF-β1RI and TGF-β1RII (panels (**A**,**B**)). Quantitative analysis of protein immunoreactivity in heart tissues (panels (**C**,**D**)). Representative immunohistochemistry of pSMAD-2 protein immunoreactivity in heart tissues (panel (**E**)). Quantitative analysis of densitometry of pSAMD-2 (panel (**F**)). Values are expressed as means ± S.E. (n = 6 animals in each group). Statistical significance is expressed as ### *p* < 0.001, vehicle-treated 0-copy and 1-copy vs. 2-copy; * *p* < 0.05, ** *p* < 0.01, vehicle-treated 0-copy,1-copy and 2 copy vs. drug-treated same gene copy number.

**Figure 6 ijms-23-11487-f006:**
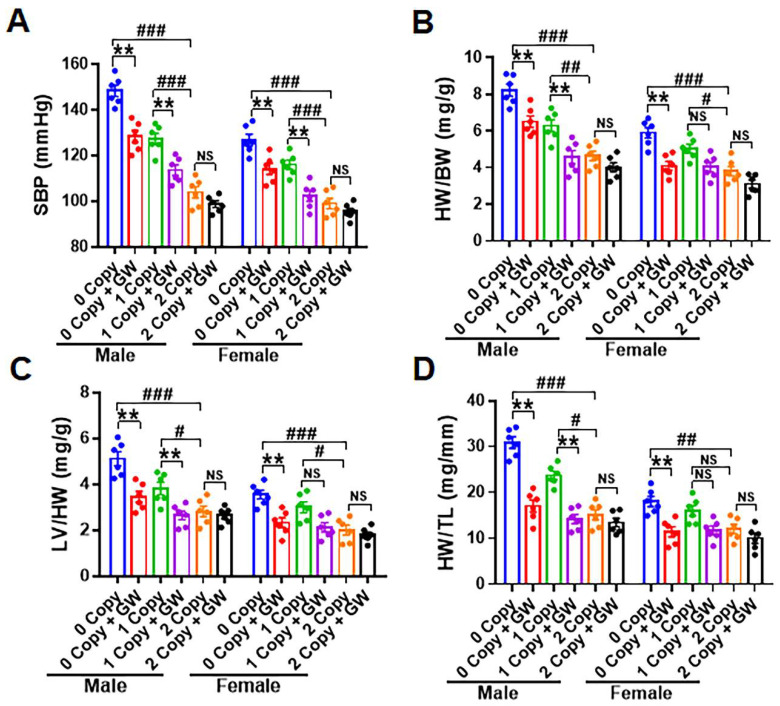
Sex-based analysis of the systolic blood pressure and cardiac hypertrophy in *Npr1* 0-copy, 1-copy, and wild-type 2-copy male and female control and GW788388-treated mice: the SBP, HW/BW, LV/BW, and TL/HW ratios are shown in panels (**A**–**D**). Values are expressed as means ± S.E. (n = 6 animals in each group). Statistical significance is expressed as # *p* < 0.05, ## *p* < 0.01, ### *p* < 0.001, vehicle-treated 0-copy and 1-copy vs. 2-copy; ** *p* < 0.01, vehicle-treated 0-copy, 1-copy and 2 copy vs. drug-treated same gene copy number. NS: not significant.

**Figure 7 ijms-23-11487-f007:**
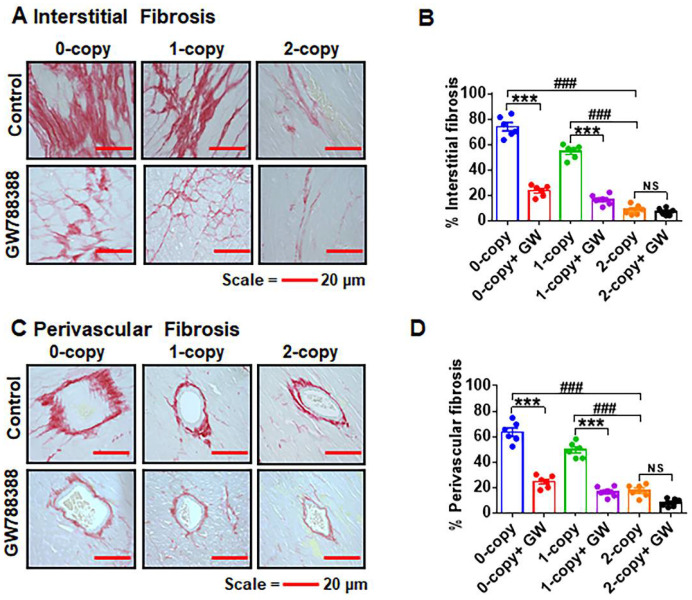
Analysis of interstitial and perivascular fibrosis in the control and drug-treated male *Npr1* gene-ablated mouse heart: Representative heart tissue sections for interstitial fibrosis from control and drug-treated male mice were stained with picrosirius red staining (panels (**A**,**B**)). The red color is indicative of fibrosis. Similarly, the representative heart tissue sections for perivascular fibrosis are presented from control and drug-treated mice (panels (**C**,**D**)). Values are expressed as means ± S.E. (n = 6 animals in each group). Statistical significance is expressed as ### *p* < 0.001, vehicle-treated 0-copy and 1-copy vs. 2-copy; *** *p* < 0.001, vehicle-treated 0-copy,1-copy and 2 copy vs. drug-treated same gene copy number. NS: not significant.

**Figure 8 ijms-23-11487-f008:**
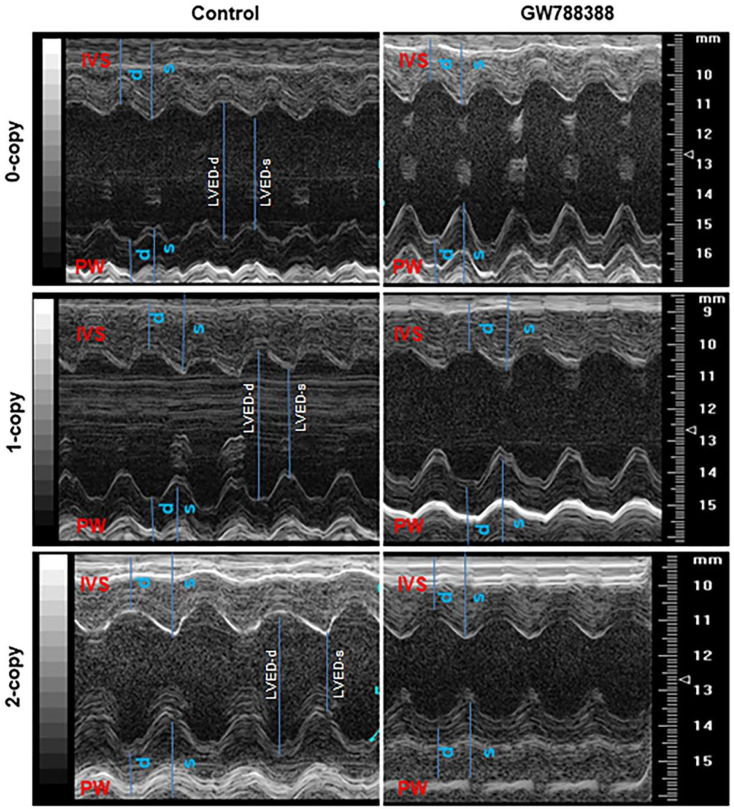
Representative images of the echocardiogram in the left ventricle of control and GW788388 drug-treated male *Npr1* gene-disrupted mice: representative M-mode images of the left ventricle for each group were calculated from the average of 6 sessions/day for five consecutive days. The representative images are shown for 0-copy, 1-copy, and 2-copy mice with and without GW788388 treatments. Longer blue lines indicate LVED-d and shorter blue lines indicate LVED-s in the echocardiogram images. Each image is the representative of a minimum of three cardiac cycles per animal (n = 6 mice per groups).

**Figure 9 ijms-23-11487-f009:**
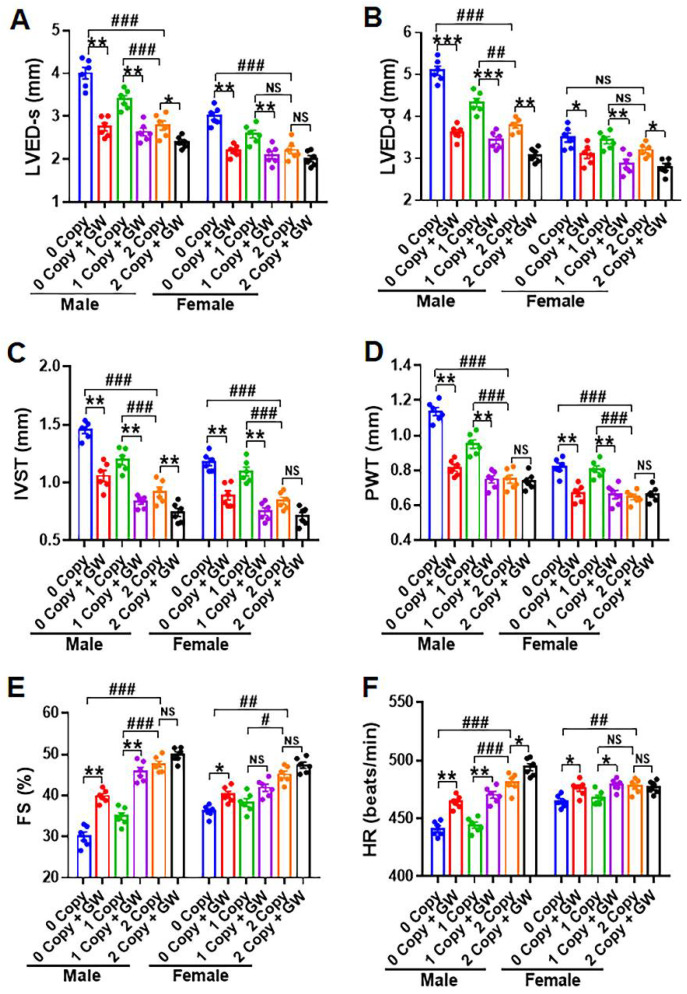
Sex-based echocardiographic analyses of cardiac structures and functions in male and female *Npr1* mutant and control mice: left ventricular end systolic dimension (LVED-s); left ventricular end diastolic dimension (LVED-d); interventricular septal wall thickness (IVST); posterior wall thickness (PWT); fractional shortening (FS); and heart rate (HR) are presented from male and female mice (panels (**A**–**F**)). Values are expressed as means ± S.E. (n = 6 animals in each group). Statistical significance is expressed as # *p* < 0.05, ## *p* < 0.01, ### *p* < 0.001, vehicle-treated 0-copy and 1-copy vs. 2-copy; * *p* < 0.05, ** *p* < 0.01, *** *p* < 0.001, vehicle-treated 0-copy, 1-copy and 2-copy vs. drug-treated same gene copy number. NS: not significant.

**Figure 10 ijms-23-11487-f010:**
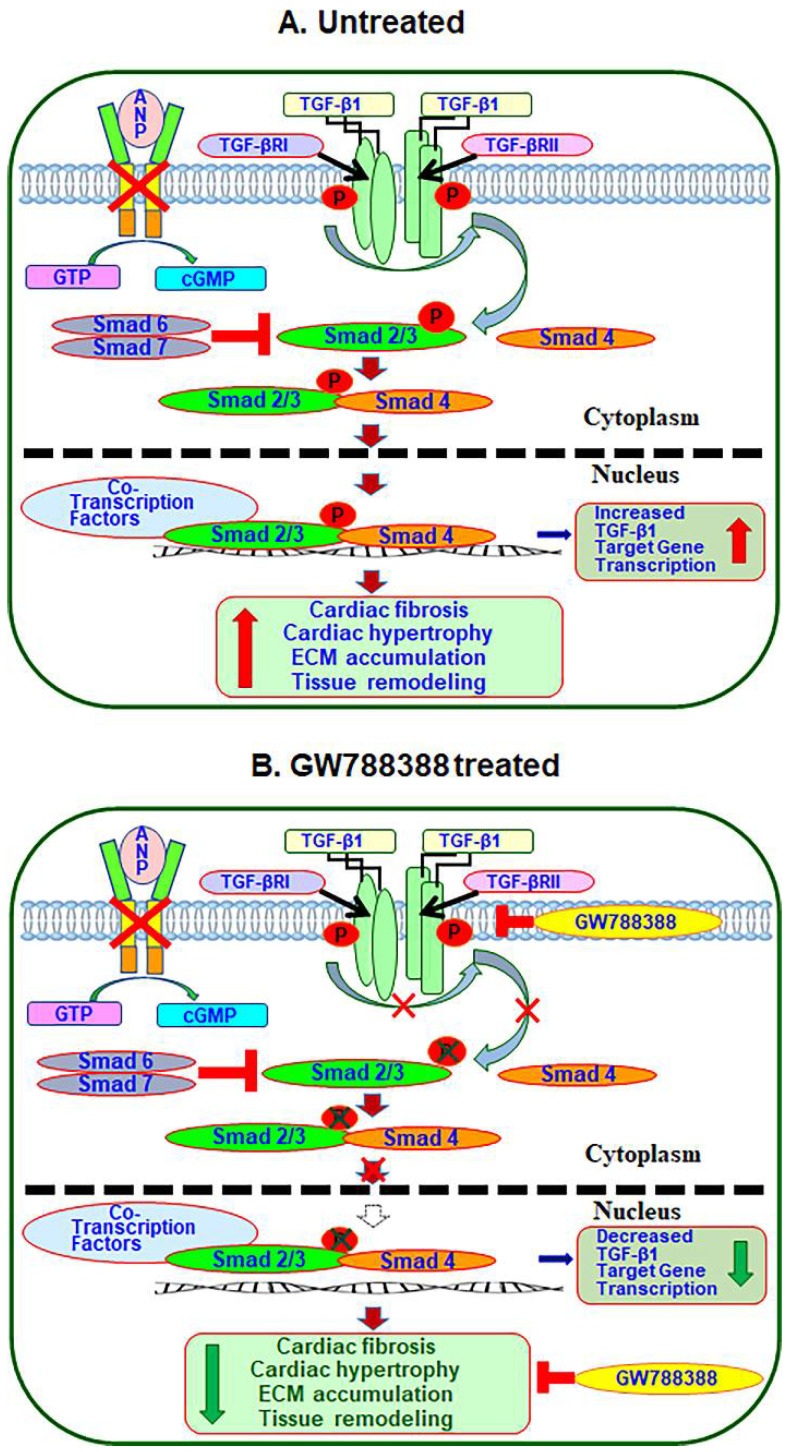
Schematic diagram representation of the proposed mechanisms whereby treatment with the TGF-β1R antagonist GW788388 inhibits the development of cardiac fibrosis and remodeling in *Npr1* gene-ablated mutant mice: disruption of *Npr1* gene leads to an unbalanced activation of TGF-β1 signaling cascade that triggers the expression of fibrotic markers, thereby activating the specific molecular and structural changes leading to fibrosis and hypertrophic remodeling in a mutant mouse heart. Treatment with the TGF-β1 receptor antagonist GW788388 ameliorates by blocking the activation and nuclear translocation of SMAD-mediated signaling networks, which protects against fibrosis and hypertrophic remodeling in 0-copy and 1-copy mutant mice hearts.

**Table 1 ijms-23-11487-t001:** List of antibodies used in the present studies, which are indicated with their application, catalog number, respective dilutions, and names of vendors of commercial sources.

Primary Antibody	Application	Catalog No.	Dilutions	Company
α-SMA	WB	sc-32251	1:250	Santa Cruz Biotechnology
CTGF	WB	sc-365970	1:300	Santa Cruz Biotechnology
ERK 1	WB	sc-271269	1:200	Santa Cruz Biotechnology
ERK 2	WB	sc-1647	1:250	Santa Cruz Biotechnology
ERK1/2	WB	sc-514302	1:300	Santa Cruz Biotechnology
MMP-2	WB	sc-13595	1:300	Santa Cruz Biotechnology
MMP-9	WB	sc-393859	1:300	Santa Cruz Biotechnology
PCNA	WB	sc-25280	1:250	Santa Cruz Biotechnology
pERK1/2	WB	sc-81492	1:250	Santa Cruz Biotechnology
pSMAD2	WB	#3108	1:250	Cell Signaling Technology
pSMAD3	WB	#9520	1:300	Cell Signaling Technology
SMAD-1	WB	#9743	1:400	Cell Signaling Technology
SMAD-2	WB	#5339	1:300	Cell Signaling Technology
SMAD-3	WB	#9513	1:250	Cell Signaling Technology
SMAD-4	WB	#46535	1:250	Cell Signaling Technology
SMAD-6	WB	#9519	1:250	Cell Signaling Technology
TGF-β1	WB	sc-130348	1:350	Santa Cruz Biotechnology
TGF-β1R1	WB	sc-101574	1:250	Santa Cruz Biotechnology
TGF-β1RII	WB	sc-17792	1:250	Santa Cruz Biotechnology
TAK1	WB	#4505	1:250	Cell Signaling Technology
pTAK1	WB	#4536	1:250	Cell Signaling Technology
Goat anti-mouse IgG2a-HRP	WB	sc-2061	1:5000	Santa Cruz Biotechnology
Goat anti-rabbit				
IgG-HRP	WB	sc-2004	1:5000	Santa Cruz Biotechnology

## Data Availability

The corresponding author has all the data which are available upon request.

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
