# Peer review of "Genetic Disruption of Guanylyl Cyclase/Natriuretic Peptide Receptor-A Triggers Differential Cardiac Fibrosis and Disorders in Male and Female Mutant Mice: Role of TGF-β1/SMAD Signaling Pathway"

_ijms, 2022, doi:10.3390/ijms231911487_

Round 1
Reviewer 1 Report
1. please mention about MMP and its full name in the abstract.
2. the figures should be in included in more quality (DPI).
3. all abbreviations have to explained when they were at the first time.
4. please consider to cite https://doi.org/10.1155/2020/3821279
5. could you updated some refs?
Author Response
RESPONSE TO THE COMMENTS FROM THE REVIEWER 1
REVIEWER 1
We appreciate the reviewer’s valuable time to review and provide the valuable comments on our manuscript, which have greatly helped to improve the contents of this manuscript.
Comments 1: Please mention about MMP and full name in abstracted.
Response: As requested by the reviewer, the MMPs (MMP-2 and MMP-9) have been included and full forms have been provided in the “Abstract” section (Page 2, lines 13, 14).
Comments 2: The Figures should be in included in more quality (DPI).
Response: As suggested by the reviewer, we have tried to improve the quality (DPI) of the Figures included in the manuscript.
Comments 3: All abbreviations have to explained when they were at the first time.
Response: As suggested by the reviewers, we have expanded all the abbreviations when they appeared at the first time, with one exception (eg., SMAD) only in the “Abstract” section due to word limit in this section of the manuscript.
Comments 4: Please consider to cite https://doi.org/10.1155/2020/3821279
Response: As recommended by the reviewer (Mediators of Inflammation, 2020; Article ID 3821279: 1-11) has been cited in the manuscript.
Comments 5: Could you updated some refs?
Response: As suggested by the reviewer, we have included some additional important references on TGF-β1 and its significance in the cellular function. A brief description is included in the revised manuscript (Page 4, lines 8-13).
Reviewer 2 Report
Subramanian et al. are attempting to elucidate sex-dependent differences in cardiac fibrosis and cardiac function using Npr1 transgenic mice. It is very interesting and has the potential to provide good information to readers of this journal, but the reasons for the sex differences are not discussed.
Furthermore, there are statistical errors and discrepancies in the interpretation of the results, as well as typographical errors and omissions, and it is doubtful that all authors have really read this paper.
Therefore, this paper is not at the level published by the IJMS.
Major points
1. The discussion is too long. The data from this paper should be properly discussed. Please delete unnecessary parts.
2. There is a gender difference between males and females, but no discussion of why this is the case
3. Why is HR decreased in this model since there is a compensatory increase in HR when contractility is decreased?
4. There is no discussion of ERK phosphorylation
5. TGFβ also acts on cardiomyocytes, but there is no analysis or discussion of the data.
6. Show representative images of the echocardiogram
7. It is very important, but the abstract mentions orally and ip, which is correct?
8. Describe the model of echocardiography
9. Provide a diagram of a representative echocardiogram
10. In the figures, there are so many statistical and interpretation errors that it is very difficult to accept. In particular, there is no statistical evaluation between Npr1-/- and Npr1+/+. Please be sure to apply the statistics.
11. In Fig.1, I don't know how many times is multiplied Npr1-/- and Npr1+/+, respectively. Also, you say 3x, etc., but in Figure 5 you say 2.5x and 1.8x, so please list to one decimal place.
12. In the last sentence in Fig. 1, almost all mRNA levels in all three groups were reduced by GW788388 treatment. This is not a decrease compared to 2-copy wild-type mice.
13. In Fig. 2, GW788388 treatment group is missing.
14. In Fig. 3, I don't know how many fold increases in Npr1-/- and Npr1+/-, respectively. Also, you say 3x, etc., please list to the first decimal place.
15. In Figure 3, "Densitometry analysis showed that the quantitative density of these fibrotic markers was significantly increased in 0-copy and 1-copy mice compared to 2-copy mice," and "The protein levels of ECM molecules, including MMP-2 (3-fold, p < 0.01) and MMP-9 (3-fold, p < 0.01) and MMP-9 (3-fold, p < 0.01), along with fibrotic marker proteins proliferating cell nuclear antigen (PCNA; 3-fold, p < 0.01), CTGF (2-fold, p < 0.01), and α-SMA (3-fold, p < 0.01), were significantly increased in both 0-copy and 1-copy mutant mice compared with 2-copy wild-type mice (Figure 3 A-E)." are duplicated
16. Figure 3: As the word "significantly" is used, "Figure 3 A-J" should be used instead of "Figure 3 A-E".
17. “treatment with GW788388 significantly reduced the quantitative levels of fibrotic proteins in both 0-copy and 1-copy mice compared to untreated control animals" Please correct it as it is reduced in all three groups.
18. Figure 4 "Furthermore, expression of TGF-β1RI (3-fold, p < 0.01) and TGF-β1RII (3-fold, p < 0.01) were also significantly increased in 0-copy"
19. I don't know how many times the number of 0- and 1-copy mice is increased in 0- and 1-copy mice, respectively. Please indicate up to one decimal place.
20. Please use (Figure 4 A-D) instead of (Figure 4 A, B). (Figure 4 A-D), not (Figure 4 C, D). (Figure 4 E, F), not (Figure 4 E-H). (Figure 4 G, H), not (Figure 4 E-H). 20.
21. In Figure 4, "The densitometry analysis showed that pSMAD2 and pSMAD3 levels were upregulated by almost 3-fold (p<001) in Npr1 The phrase "gene-disrupted 0-copy mice compared with 2-copy mice" should be corrected to 0-copy mice and 1-copy mice instead of 0-copy mice.
22. P8 2.3. In the fifth and fourth lines from the bottom, the word "significantly" is used, but compared to what?
23. p8 2.3. the third line from the bottom shows "in Npr1-/- mice", but there is a significant difference in other mice.
24. p8 2.4. "The SBP in 0-copy male mice was increased to 146 + 5 mmHg and 128 + 3 mmHg in 1-copy male mice, compared with 101 + 2 mmHg The SBP in 0-copy male mice was increased to 146 + 5 mmHg and 128 + 3 mmHg in 1-copy male mice, compared with 101 + 2 mmHg in 2-copy wild-type male mice. English is something strange.
25. In Figure 6A, SBP should be stated that there was no effect of GW788388 in WT
26. p8 2.4. "Further, it was observed that the HW/BW ratio was significantly increased by almost 60% (global hypertrophy) in adult 0-copy male mice (8.02 + 0.71) and 48% in 1-copy male mice (5.92 + 0.64) compared with 2-copy wild-type male mice (4.35 + 0.42). -copy male mice (4.35 + 0.42)", but there is no significant difference in 1-copy male mice. Also, the value of 5.92 + 0.64 is not correct because it is greater than 6 on the graph.
27. P8 2.4. "However, the HW/BW ratio was not significantly" but there is a significant difference.
28. P8 2.4, "treatment with GW788388 almost normalized the HW/BW ratio in both 0-copy and 1-copy male mice". However, it does not appear to be normalized in 0-copy male mice.
29. P8 2.4. "Significant increases were also observed in the ratio of LVW/BW in both 0-copy and 1-copy male mice"
30. But, significant increases were also observed in the ratio of LVW/BW in both 0-copy and 1-copy male mice
31. P8 2.4. "The LVW/BW ratio was not significantly altered in either 0-copy or 1-copy female mice.
32. But there is a significant difference in 0-copy female mice.
33. P8 2.4. "The TL/BW ratio was significantly increased in 0-copy male mice (32.8 + 1.2) and 1-copy male mice (23 + 0.8) compared with 2-copy male mice (18 + 0.5)."
34. However, there is no significant difference in 1-copy male mice.
35. p8 2.4. "Treatment with GW788388 normalized the TL/BW ratio to the levels observed in control wild-type mice", but it does not appear to be normalized from the graph.
36. p8 2.4. "However, the TL/BW ratio in 0-copy and 1-copy female mice was not significantly altered and remained at almost However, the TL/BW ratio in 0-copy and 1-copy female mice was not significantly altered and remained at almost
37. There is a significant difference.
38. “Importantly, the HW/BW, LV/BW, and HW/TL ratios were significantly reduced (p<0.01) satisfactorily above the normal values in Npr1-/- mice treated with GW788388.”
39. The meaning is not clear.
40. In Fig. 5E, it is difficult to detect the positive ones compared to A and B.
41. P10 2.5. ed progressive interstitial cardiac fibrosis, with an abrupt increase in the deposition of interstitial collagen fibers (Figure 7 A, B).” It is difficult to understand what it says.
42. p10 2.5. “Male Npr1 0-copy and 1-copy mutant mice treated with GW788388 showed attenuated levels of interstitial cardiac fibrosis (40 - 45%, p<0.001) compared to wild-type 2-copy control male mice”, isn't "compared to wild-type 2-copy control male mice" strange? Isn't the comparison to wild-type 2-copy control male mice wrong? Also, the 40 - 45% rate is the same for 0-copy and 1-copy mutant mice, but how much for each? Please indicate to one decimal place.
43. P10 2.5. "Moreover, no abnormal accumulation of collagen fibers was noted in control or GW788388-treated animals. What is "animals"?
44. P10 2.5.(20 - 25%), what is the percentage of 0-copy and 1-copy mutant mice, respectively? Please indicate to one decimal place.
45. P11 2.6. "LVED-d was also significantly increased in both 0-copy and 1-copy male mice, and after treatment with GW788388, the values were normalized to control levels”. "control levels" should be "2-copy control male mice.
46. P11 2.6. There are three 1-copy male mice, but they are not significantly different.
Minor points
1. large font such as Npr1 in intro, please unify
2. p3 2.1 line 7 from the top ➡, P<0.001). TGF-β1R1 receptors ➡ "." should be ",".
3. In the figure, the thickness of the line under # is different.
4. In Fig. 1, G and H should be reversed.
5. fig. 2, which WB was performed using the nuclear fraction or the cytosolic fraction, respectively?
6. Which fraction was used for WB for total TAK1? 7.
7. Figure 2H: "Corrected for total ERK," but I think it was probably corrected for β-actin.
8. Fig. 3 Representative western blots and Densitometry analysis do not match.
9. figure 3 Legend (panels A-H) should be corrected to (panels A-E).
10. p8 2.4, 3rd line from the top, "cardiac dysfunction analyses (Figure 6 A-D)" should be deleted because it is a mistake for Figure 8.
11. P8 2.4, TL/BW should be HW/TL.
12. P8 2.4 "Discovery" at the end is unnecessary.
13. Fig. 5C Vertical axis %Positive area for" is the number of positive cells written in the method section.
14. What are you comparing "*"to?
15. figures 7b and d, how are they quantified? 16.
16. fig.7B Is there a significant difference between 2-copy and 2-copy +GW?
17. p8 2.6. "However," is duplicated.
18. Figure 8, what are you comparing "*" to?
Author Response
RESPONSE TO THE COMMENTS FROM THE REVIEWER 2
REVIEWER 2
We appreciate the reviewer’s valuable time to review and provide the constructive comments on our manuscript, which have helped to improve the contents of this manuscript to a great extent. We acknowledge the reviewer’s comments that the manuscript is interesting and has potential to provide good information to readers of this journal.
Major points
Comments 1: The discussion is too long. The data from this paper should be properly discussed. Please delete unnecessary parts.
Response: As indicated by the reviewer, the “Discussion” section has been shortened and the findings have been presented in more systematic fashion.
Comments 2: There is a gender difference between males and females, but no discussion of why this is the case
Response: As suggested by the reviewer the gender difference between males and females has been more carefully stated in the revised manuscript (Page 5, lines 11-17; Page 15, lines 16-19).
Comments 3: Why is HR decreased in this model since there is a compensatory increase in HR when contractility is decreased?
Response: we appreciate the reviewer’s comments. We agree that the heart rate (HR) was decreased in Npr1 0-copy and 1-copy mutant mice. After approximately 6 months of age, 0-copy male mutant mice die due to congestive heart failure (CHF). The heart seems to be very much deteriorated and the compensatory mechanisms are compromised and do not seem to operate and maintain the heart function and HR in these animals. A brief statement has also been incorporated in the revised manuscript (Page 15, lines 15-20).
Comments 4: There is no discussion of ERK phosphorylation
Response: ERK phosphorylation (pERK1/2) has been stated in the revised manuscript (Page 7, lines 11, 12).
Comments 5: TGFβ also acts on cardiomyocytes, but there is no analysis or discussion of the data.
Response: We studied the TGF-β1 signaling in the cardiac tissues isolated from the intact animals in vivo. We did not examine the isolated cardiomyocytes in culture condition. In future, we expect to continue our ongoing studies and plan to examine the effect of TGF-β1 in the isolated cardiomyocytes; however, the results of those future studies will be reported independently in a separate manuscript.
Comments 6: Show representative images of the echocardiogram
Response: As requested by the reviewer, we have included the Figure 8, representative M-mode echocardiogram images of the left ventricle for each group in the revised manuscript.
Comments 7: It is very important, but the abstract mentions orally and ip, which is correct?
Response: We apologize for this inadvertent error. In the present studies, ip method was used to deliver TGF-β1 in mice. However, in the preliminary studies both ip. and oral methods were used for comparison purposes. There was no significant difference in both methods, therefore, ip method was adopted in current studies. The inadvertent error “orally” has been deleted in the “Abstract” section of the revised manuscript (Page 2, lines 5, 6).
Comments 8: Describe the model of echocardiography
Response: In response to the reviewer’s suggestions, we have included the description of the model of echocardiography in the Material and Methods section of the revised manuscript.
Cardiac structure and function were assessed via echocardiography using the Vevo 3100 Imaging System with a 30-MHz probe (VisualSonics, Toronto, Canada) as previously described. Mice were anesthetized with 1 to 1.5% isoflurane and placed onto an integrated heating pad to maintain body temperature. B-mode and M-mode images of the long and short axis were recorded. Images of the left ventricular chamber were analyzed using the leading-edge method in Vevo LAB 5.5.1 Software (VisualSonics, Toronto, Canada). Group averages were calculated from measurements made from a minimum of three cardiac cycles per animal. Fractional shortening (FS); left ventricular end-diastolic dimension (LVED-d); left ventricular end-systolic dimension (LVES-d); interventricular septal wall thickness diastolic (IVST-d); and posterior wall thickness diastolic (PWT-d) were calculated. A brief description has been incorporated in the revised manuscript (Page 19, lines 19-23; page 20, lines 1-4).
Comments 9: Provide a diagram of a representative echocardiogram
Response: We have included the representative images of echocardiogram in Figure 8 of the revised manuscript (Please see our response to comment # 6).
Comments 10: In the figures, there are so many statistical and interpretation errors that it is very difficult to accept. In particular, there is no statistical evaluation between Npr1-/- and Npr1+/+. Please be sure to apply the statistics.
Response: As recommended by the reviewer, we have reexamined and reevaluated all the statistical significance and the interpretation throughout the “Results” section of this manuscript. We have included the comparisons between Npr1-/- and Npr1+/+ along with GW788388-treated and untreated controls. Now, we have applied more carefully the statistical comparisons in the revised manuscript (Figures 1-7, Figure 8).
Comment 11: In Fig.1, I don't know how many times is multiplied Npr1-/- and Npr1+/+, respectively. Also, you say 3x, etc., but in Figure 5 you say 2.5x and 1.8x, so please list to one decimal place.
Response: As pointed by the reviewer, in the Figure 1, now we have corrected the fold changes to at least one decimal place. The corrections have been incorporated in the revised manuscript (Page 6, lines 8-15).
Comment 12: In the last sentence in Fig. 1, almost all mRNA levels in all three groups were reduced by GW788388 treatment. This is not a decrease compared to 2-copy wild-type mice.
Response: As suggested by the reviewer, the phrase “almost all” has been deleted. The statement has been added indicating “GW788388-mediated inhibition of mRNA was still greater in 0-copy and 1-copy mice than untreated control levels”.
Comment 13: In Fig. 2, GW788388 treatment group is missing.
Response: We appreciate the reviewer’s comment. However, due to a very large number of treatment and control sample sizes, we were unable to include GW788388-treated groups in the Figure 2 Western blot analyses. We largely relied on mRNA data in Figure 1 to evaluate and project the differences in the expression levels of these marker proteins.
Comment 14: In Fig. 3, I don't know how many fold increases in Npr1-/- and Npr1+/-, respectively. Also, you say 3x, etc., please list to the first decimal place.
Response: As suggested by the reviewer, in Figure 3, the fold increases in Npr1-/- and Npr1+/- mice have been corrected in the text in correspondence to the Figure bars, respectively, to at least one decimal place (Page 7, lines 17-21).
Comment 15: In Figure 3, "Densitometry analysis showed that the quantitative density of these fibrotic markers was significantly increased in 0-copy and 1-copy mice compared to 2-copy mice," and "The protein levels of ECM molecules, including MMP-2 (3-fold, p < 0.01) and MMP-9 (3-fold, p < 0.01) and MMP-9 (3-fold, p < 0.01), along with fibrotic marker proteins proliferating cell nuclear antigen (PCNA; 3-fold, p < 0.01), CTGF (2-fold, p < 0.01), and α-SMA (3-fold, p < 0.01), were significantly increased in both 0-copy and 1-copy mutant mice compared with 2-copy wild-type mice (Figure 3 A-E)." are duplicated
Response: As suggested by the reviewers, the duplication in the description of densitometry analysis in Figure 3 has been corrected in the revised manuscript (Page 7, lines 17-21; Please see response to comment #14).
Comment 16: Figure 3: As the word "significantly" is used, "Figure 3 A-J" should be used instead of "Figure 3 A-E".
Response: As suggested by the reviewer, Figure 3A-J has been used instead of Figure 3 A-E in the revised manuscript (Page 7, line 22).
Comment 17: “treatment with GW788388 significantly reduced the quantitative levels of fibrotic proteins in both 0-copy and 1-copy mice compared to untreated control animals" Please correct it as it is reduced in all three groups.
Response: As indicated by the reviewer, the correction has been made to indicate that GW788388 reduced the fibrotic protein levels in all three groups (Page 7, lines 22; page 8, lines 1, 2).
Comment 18: Figure 4 "Furthermore, expression of TGF-β1RI (3-fold, p < 0.01) and TGF-β1RII (3-fold, p < 0.01) were also significantly increased in 0-copy"
Response: As suggested by the reviewer, the fold values for TGF-β1RI and TGF-β1RII has been corrected to at least one decimal place (Page 8, lines 20, 21; Page 9, lines 1, 2).
Comment 19: I don't know how many times the number of 0- and 1-copy mice is increased in 0- and 1-copy mice, respectively. Please indicate up to one decimal place.
Response: As suggested by the reviewer the fold changes have been presented to one decimal place in the revised manuscript (Page 9, lines 5-7).
Comment 20: Please use (Figure 4 A-D) instead of (Figure 4 A, B). (Figure 4 A-D), not (Figure 4 C, D). (Figure 4 E, F), not (Figure 4 E-H). (Figure 4 G, H), not (Figure 4 E-H). 20.
Response: In accordance with the reviewer’s comments, the Figure 4 A-D is used instead Figure 4 A, B, Figure 4 A-D, not (Figure 4 C, D). Similarly, (Figure 4 E, F) not (Figure 4 E-H), and Figure 4 G, H, not Figure 4 E-H (Page 8, lines 8-14).
Comment 21: In Figure 4, "The densitometry analysis showed that pSMAD2 and pSMAD3 levels were upregulated by almost 3-fold (p<001) in Npr1 The phrase "gene-disrupted 0-copy mice compared with 2-copy mice" should be corrected to 0-copy mice and 1-copy mice instead of 0-copy mice.
Response: As suggested by the reviewer, the densitometry analysis of pSMAD2 and pSMAD3 has been corrected to fold value with one decimal places. Also, the phrase gene-disrupted 0-copy mice and 1-copy mice instead of 0-copy mice (Page 8, lines 11-14).
Comment 22: P8 2.3. In the fifth and fourth lines from the bottom, the word "significantly" is used, but compared to what?
Response: As indicated by the reviewer, the correction has been made to indicate “was significantly increased in Npr1-/- mice compared with 2-copy mice (Page 9, lines 1, 2).
Comment 23: p8 2.3. the third line from the bottom shows "in Npr1-/- mice", but there is a significant difference in other mice.
Response: As indicated by the reviewer, the statement that the significant differences were also observed in other mice (2-copy) mice in the revised manuscript (Page 9, lines 8, 9).
Comment 24: p8 2.4. "The SBP in 0-copy male mice was increased to 146 + 5 mmHg and 128 + 3 mmHg in 1-copy male mice, compared with 101 + 2 mmHg The SBP in 0-copy male mice was increased to 146 + 5 mmHg and 128 + 3 mmHg in 1-copy male mice, compared with 101 + 2 mmHg in 2-copy wild-type male mice. English is something strange.
Response: As suggested by the reviewer, the sentence structure has been corrected and changed to clearly state the results of SBP in Npr1 mice (Page 9, lines 17-22).
Comment 25: In Figure 6A, SBP should be stated that there was no effect of GW788388 in WT
Response: It has been stated that there was no significant effect of GW788388 on the SBP in 2-copy WT mice (Page 10, lines 1-4).
Comment 26: p8 2.4. "Further, it was observed that the HW/BW ratio was significantly increased by almost 60% (global hypertrophy) in adult 0-copy male mice (8.02 + 0.71) and 48% in 1-copy male mice (5.92 + 0.64) compared with 2-copy wild-type male mice (4.35 + 0.42). -copy male mice (4.35 + 0.42)", but there is no significant difference in 1-copy male mice.
Response: We appreciate the reviewer’s comment. Now, we have corrected the significant differences in 1-copy male mice. In addition, the value 5.92+0.64 has been corrected to 6.4+0.65 in the revised manuscript (Page 10, lines 4-7).
Comment 27: P8 2.4. "However, the HW/BW ratio was not significantly" but there is a significant difference.
Response: The significance difference for HW/BW ratio in female mice has been corrected in the revised manuscript (Page 10, lines 7, 8).
Comment 28: P8 2.4, "treatment with GW788388 almost normalized the HW/BW ratio in both 0-copy and 1-copy male mice". However, it does not appear to be normalized in 0-copy male mice.
Response: The treatment with GW788388 did not normalize HW/BW ratio in 0-copy male mice, but almost normalized HW/BW ratio in 1-copy male mice (Page 10, lines 8-10).
Comment 29: P8 2.4. "Significant increases were also observed in the ratio of LVW/BW in both 0-copy and 1-copy male mice"
Response: “significant increases were observed in LVW/BW ratio of 0-copy female mice” has been corrected in the revised manuscript (Page 10, lines 10-13).
Comment 30: But, significant increases were also observed in the ratio of LVW/BW in both 0-copy and 1-copy male mice
Response: Please see our response as indicated in “Response” to comment # 29.
Comment 31: P8 2.4. "The LVW/BW ratio was not significantly altered in either 0-copy or 1-copy female mice.
Response: We have corrected the LVW/BW ratio that was significantly altered in 0-copy female mice and the correction has been incorporated in the revised manuscript (Page 10, lines 13, 14).
Comment 32: But there is a significant difference in 0-copy female mice.
Response: Please see our response as indicated in “Response” to comment # 31.
Comment 33: P8 2.4. "The TL/BW ratio was significantly increased in 0-copy male mice (32.8 + 1.2) and 1-copy male mice (23 + 0.8) compared with 2-copy male mice (18 + 0.5)."
Response: As suggested by the reviewer, the values have been corrected to one decimal place in the revised manuscript (Page 10, line 10; lines 16-18).
Comment 34: However, there is no significant difference in 1-copy male mice.
Response: The significant difference has been corrected to at least one decimal place in the revised manuscript (Page 10, lines 17, 18).
Comment 35: p8 2.4. "Treatment with GW788388 normalized the TL/BW ratio to the levels observed in control wild-type mice", but it does not appear to be normalized from the graph.
Response: The statement “treatment with GW788388 did not normalize TL/BW ratio to the levels in control WT mice has been incorporated in the revised manuscript (Page 10, lines 18, 19).
Comment 36: p8 2.4. "However, the TL/BW ratio in 0-copy and 1-copy female mice was not significantly altered and remained at almost However, the TL/BW ratio in 0-copy and 1-copy female mice was not significantly altered and remained at almost.
Response: The statement has been modified “The TL/BW ratio was significantly altered only in 0-copy female mice, but in 1-copy female mice remained at almost similar levels to 2-copy control WT mice (Page 10, lines 19, 20).
Comment 37: There is a significant difference.
Response: Correction has been incorporated in the revised manuscript to indicate significant differences (page 10, lines 20, 21).
Comment 38: “Importantly, the HW/BW, LV/BW, and HW/TL ratios were significantly reduced (p<0.01) satisfactorily above the normal values in Npr1-/- mice treated with GW788388.”
Response: For clarity purposes, the sentence has been deleted in the revised manuscript.
Comment 39: The meaning is not clear.
Response: As above, the sentence has been deleted in the revised manuscript.
Comment 40: In Fig. 5E, it is difficult to detect the positive ones compared to A and B.
Response: Usually, SMAD proteins are rapidly degraded during the tissue processing for immunostaining. The antibodies specificity is also an important factor to detect the signal, which is variable. These factors seem to have contributed for weaker positive signals.
Comment 41: P10 2.5. ed progressive interstitial cardiac fibrosis, with an abrupt increase in the deposition of interstitial collagen fibers (Figure 7 A, B).” It is difficult to understand what it says.
Response: The sentence has been revised to make it grammatically correct in the revised manuscript (Page 11, lines 2-6).
Comment 42: p10 2.5. “Male Npr1 0-copy and 1-copy mutant mice treated with GW788388 showed attenuated levels of interstitial cardiac fibrosis (40 - 45%, p<0.001) compared to wild-type 2-copy control male mice”, isn't "compared to wild-type 2-copy control male mice" strange? Isn't the comparison to wild-type 2-copy control male mice wrong? Also, the 40 - 45% rate is the same for 0-copy and 1-copy mutant mice, but how much for each? Please indicate to one decimal place.
Response: The sentence has been revised to indicate the appropriate comparisons “untreated control” instead 2-copy control mice. Male Npr1 0-copy mice showed 65-70% fibrosis and 1-copy male showed 40-45% cardiac fibrosis (Page 11, lines 7-12).
Comment 43: P10 2.5. "Moreover, no abnormal accumulation of collagen fibers was noted in control or GW788388-treated animals. What is "animals"?
Response: The genotype of animal has been provided as “2-copy male mice” in revised manuscript (Page 10, lines 11-13).
Comment 44: P10 2.5.(20 - 25%), what is the percentage of 0-copy and 1-copy mutant mice, respectively? Please indicate to one decimal place.
Response: In both 0-copy and 1-copy mice the percentage values have been used with the range values (Page 11, lines 11-13).
Comment 45: P11 2.6. "LVED-d was also significantly increased in both 0-copy and 1-copy male mice, and after treatment with GW788388, the values were normalized to control levels”. "control levels" should be "2-copy control male mice.
Response: As indicated by the reviewer, for the control levels, 2-copy control mice have been included in the revised manuscript (Page 12, lines 2-4).
Comment 46: P11 2.6. There are three 1-copy male mice, but they are not significantly different.
Response: Somehow, we are not able to point to the exact critique indicating “there are three 1-copy male mice, we are unable to find the error in the original manuscript.
Minor Points:
Comment 1: large font such as Npr1 in intro, please unify
Response: “Npr1” is written according to the gene nomenclature; first letter is written in uppercase followed by lowercase letters and numbers, all in italic for eg. Npr1. For protein nomenclature, all letters are written in uppercase for e.g. NPRA or GC-A/NPRA.
Comment 2: p3 2.1 line 7 from the top➡, P<0.001). TGF-β1R1 receptors➡ "." should be ",".
Response: The fold values have been changed to one decimal places in the revised manuscript.
Comment 3: In the figure, the thickness of the line under # is different.
Response: As suggested by the reviewer, the thickness of the lines in Figures have been corrected in the revised manuscript.
Comment 4: In Fig. 1, G and H should be reversed.
Response: As indicated by the reviewer, in Figure 1, the panels G and H have been corrected in the revised manuscript.
Comment 5: fig. 2, which WB was performed using the nuclear fraction or the cytosolic fraction, respectively?
Response: In Figure 2, A-F and G, Western blot was done in the nuclear extract. On the other hand, in Figure 2, F and H, the Western blot was done in the cytoplasmic extract. The corrections have been incorporated in the legend to Figure 2 in the revised manuscript.
Comment 6: Which fraction was used for WB for total TAK1? 7.
Response: Inadvertently, we referred to total TAK1. The phrase “total” has been deleted and corrected in the revised manuscript. the pTAK1 protein was expressed in relation to the TAK1. The correction has been incorporated in the revised manuscript (Figure 2).
Comment 7: Figure 2H: "Corrected for total ERK," but I think it was probably corrected for β-actin.
Response: The pERK1/2 was corrected to ERK1/2. The correction has been incorporated in the revised manuscript.
Comment 8: Fig. 3 Representative western blots and Densitometry analysis do not match.
Response: In Figure 3, we have now tried to match the densitometry analysis in the manuscript.
Comment 9: figure 3 Legend (panels A-H) should be corrected to (panels A-E).
Response: In Figure 3, legend, the panels A-H has been corrected to panels A-E in the revised manuscript (Page 7, lines 17-20).
Comment 10: p8 2.4, 3rd line from the top, "cardiac dysfunction analyses (Figure 6 A-D)" should be deleted because it is a mistake for Figure 8.
Response: Revised Figure 9: The “cardiac dysfunction changes” has been corrected to cardiac mass and hypertrophy in the revised manuscript (Page 12, lines 2-9).
Comment 11: P8 2.4, TL/BW should be HW/TL.
Response: As indicated by the reviewer, TL/BW has been corrected to HW/TL in the revised manuscript (Page 10, lines 16-18).
Comment 12: P8 2.4 "Discovery" at the end is unnecessary.
Response: The word “Discovery” at the end of the sentence has been deleted to make the sentence grammatically correct.
Comment 13: Fig. 5C Vertical axis %Positive area for" is the number of positive cells written in the method section.
Response: The “number of positive cells” has been corrected to “% positive area” in the revised manuscript.
Comment 14: What are you comparing "*"to?
Response: The comparison has been corrected between 0-copy and 1-copy mice versus 2-copy mice and between drug-treated and untreated control mice.
Comment 15: figures 7b and d, how are they quantified? 16.
Response: The signals in Figure 7 b and d photomicrographs were quantified using Image-Proplus image analysis software (Page 24, lines 2-4).
Comment 16: fig.7B Is there a significant difference between 2-copy and 2-copy +GW?
Response: There is no significant difference between untreated and GW-treated 2-copy mice. The correction has been incorporated in the revised manuscript.
Comment 17: p8 2.6. "However," is duplicated.
Response: The error has been corrected by changing “However” with “furthermore” in the revised manuscript.
Comment 18: Figure 8, what are you comparing "*" to?
Response: The comparisons have been made between 0-copy and 1-copy with 2-copy WT mice. Also, comparisons have been included between drug-treated versus untreated groups in the revised manuscript (please see revised Figure 9 legends).

Round 2
Reviewer 1 Report
the paper has been improved.
Author Response
thank you for your comments.
Reviewer 2 Report
The authors responded appropriately to my comments.
Author Response
thank you for your comments.